# POLTER: Policy Trajectory Ensemble Regularization for Unsupervised Reinforcement Learning

**Frederik Schubert**                                    *schubert@tnt.uni-hannover.de*
*Institute for Information Processing*
*Leibniz University Hannover*

**Carolin Benjamins**                                    *benjamins@ai.uni-hannover.de*
*Institute of Artificial Intelligence*
*Leibniz University Hannover*

**Sebastian Döhler**                                     *doehler@tnt.uni-hannover.de*
*Institute for Information Processing*
*Leibniz University Hannover*

**Bodo Rosenhahn**                                       *rosenhahn@tnt.uni-hannover.de*
*Institute for Information Processing*
*Leibniz University Hannover*

**Marius Lindauer**                                      *m.lindauer@ai.uni-hannover.de*
*Institute of Artificial Intelligence*
*Leibniz University Hannover*

**Reviewed on OpenReview:** *https: // openreview. net/ forum? id=Hnr23knZfY*

## Abstract

The goal of Unsupervised Reinforcement Learning (URL) is to find a reward-agnostic prior policy on a task domain, such that the sample-efficiency on supervised downstream tasks is improved. Although agents initialized with such a prior policy can achieve a significantly higher reward with fewer samples when finetuned on the downstream task, it is still an open question how an optimal pretrained prior policy can be achieved in practice. In this work, we present POLTER (Policy Trajectory Ensemble Regularization) – a general method to regularize the pretraining that can be applied to any URL algorithm and is especially useful on data- and knowledge-based URL algorithms. It utilizes an ensemble of policies that are discovered during pretraining and moves the policy of the URL algorithm closer to its optimal prior. Our method is based on a theoretical framework, and we analyze its practical effects on a white-box benchmark, allowing us to study POLTER with full control. In our main experiments, we evaluate POLTER on the Unsupervised Reinforcement Learning Benchmark (URLB), which consists of 12 tasks in 3 domains. We demonstrate the generality of our approach by improving the performance of a diverse set of data- and knowledge-based URL algorithms by 19% on average and up to 40% in the best case. Under a fair comparison with tuned baselines and tuned POLTER, we establish a new state-of-the-art for model-free methods on the URLB.

## 1 Introduction

Reinforcement Learning (RL) has shown many successes in recent years (Mnih et al., 2015; Silver et al., 2017; OpenAI et al., 2019) and is starting to have an impact on real-world applications (Bellemare et al., 2020; Mirhoseini et al., 2021; Degrave et al., 2022). However, all these applications require detailed knowledge of the task to train the agents in a sufficiently close simulation. Implementing each and every task and

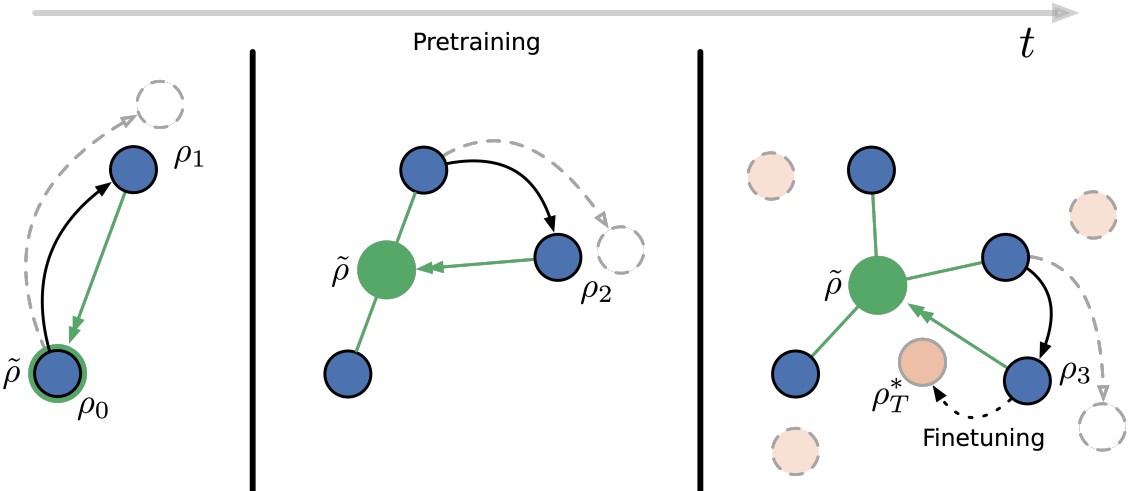

Figure 1: Optimization trajectory in the induced state-marginal space $\rho(s)$ of a policy $\pi(s)$ that optimizes an Unsupervised Reinforcement Learning objective with Policy Trajectory Ensemble Regularization (POLTER) regularization. The regularized policy's state distribution $\rho_t(s)$ ⬤ is pulled towards the ensemble state distribution $\tilde{\rho}(s)$ ⬤ during pretraining. The ensemble policy $\tilde{\pi}(s)$ approximates the optimal prior $\pi_T^*(s)$ with its induced state-marginal $\rho_T^*(s)$ ⬤ for finetuning on a specific task $T$. Note that other tasks ⬤ might lie further away from the average. However, as shown by Eysenbach et al. (2022), the average still minimizes the distance without knowledge about the specific downstream task. The state distribution trajectory of a policy without POLTER regularization is indicated by dashed lines ⬤.

training a new agent is time-consuming and inefficient. In some cases, the task might be difficult to model in simulation, or it is unknown beforehand, which makes sample-efficiency even more important. Unsupervised Reinforcement Learning (URL), i.e., reward-free pretraining in a task domain (Srinivas and Abbeel, 2021), has shown to improve the sample-efficiency of finetuning RL algorithms on downstream tasks. This setup only requires a simulation of the agent's domain without having to model task-specific components in detail. Although URL results in a higher sample-efficiency on some applications, pretraining a policy still does not guarantee an improvement in performance on a wide range of tasks. For example, Laskin et al. (2021) observed that longer pretraining often degrades the performance of many URL algorithms. Also, many finetuning steps are still needed to achieve the optimal return, and some algorithms fail to learn a sufficient policy at all.

In this work, we introduce POLTER (Policy Trajectory Ensemble Regularization) – a general method to improve the performance of URL algorithms via a novel regularization term during pretraining that results in better prior policies for finetuning on specific tasks. Our theoretical motivation builds on the geometric interpretation of URL by Eysenbach et al. (2022) who connect the optimal prior policy to a specific state distribution. The prior state distribution is optimal in the sense that it minimizes the distance to the state distribution of an adversarially chosen downstream task. This problem statement can be seen an instance of the information-theoretic framework of bounded rationality (Ortega and Lee, 2014). Some URL algorithms try to approximate the optimal prior state distribution by learning separate skills during pretraining that are combined to initialize the finetuning policy. POLTER borrows this idea for URL algorithms that do not learn explicit skills, e.g., data- and knowledge-based URL algorithms. This is based on the insight that each point during pretraining encodes an implicit skill that depends on the pretraining objective. In a nutshell, our main idea is to minimize the distance of the current policy to an ensemble policy during pretraining, which acts as a proxy for the optimal prior. As shown in Figure 1, the pretraining trajectory of the policy and hence the induced state distribution is affected by our regularization and attracted to the ensemble of earlier pretraining policies, each potentially encoding a different skill.

We demonstrate the effect of POLTER on the simplistic PointMass environment and two tasks in the Pendulum domain, and provide empirical evidence that our method reduces the distance to the optimal prior policy and

improves the downstream performance. Extensive experiments on the Unsupervised Reinforcement Learning Benchmark (URLB) (Laskin et al., 2021) show that our method improves the Interquartile Mean (IQM) (Agarwal et al., 2021) performance of data- and knowledge-based URL algorithms on average by 19%. A sensitivity analysis reveals that POLTER is a robust method with a reasonable default setting but can further be tuned on some domains and algorithms. Additional experiments on the state-visitation entropy provide empirical evidence that our method improves the performance by finding better priors in contrast to an improved exploration. In summary, our **contributions** are:

1. We introduce POLTER, a novel regularization method for URL algorithms that uses an ensemble of pretraining policies to reduce the deviation from the optimal prior policy.

2. Our method is empirically validated on the white-box PointMass benchmark and demonstrated in the Pendulum domain.

3. An extensive evaluation on the URLB shows the effectiveness of POLTER for a broad range of URL algorithms with pixel- and state-based observations.

4. With our sensitivity analysis, we demonstrate that our approach is robust with a good default and can be further boosted via a task-specific strength of the regularization and achieves a new state-of-the-art for model-free algorithms on the URLB.

## 2  Preliminaries

### 2.1  Notation in Reinforcement Learning (RL)

RL relies on the notion of a Markov Decision Process (MDP) (Sutton and Barto, 1998). An MDP is a tuple $(\mathcal{S}, \mathcal{A}, \mathcal{P}, r, \gamma)$ which describes the states $s \in \mathcal{S}$, actions $a \in \mathcal{A}$, transition dynamics $p(s_{t+1} \mid s_t, a_t) \sim \mathcal{P}$, initial state distribution $p_0(s_0) \sim \mathcal{P}$, reward function $r(s_t)$ and discount factor $\gamma \in (0, 1]$ of an environment. At each time step $t$, an agent observes the state $s_t$ of the environment and chooses an action $a_t$ using a policy $\pi(a_t \mid s_t)$ to transition into a new state $s_{t+1}$. The objective is to maximize the expected discounted return by finding a policy $\pi^*(s)$ that induces the optimal discounted state occupancy measure or state-marginal $\rho^*(s) = (1 - \gamma) \sum_{t=0}^{\infty} \gamma^t P_t^{\pi^*}(s)$, where $P_t^{\pi^*}(s)$ is the probability of being in state $s$ at time step $t$ when following policy $\pi^*$. As we focus on state-based reward functions $r(s)$, this optimal discounted state occupancy measure corresponds to the maximal discounted return of the optimal policy.

### 2.2  Related Work

**Unsupervised Reinforcement Learning**  describes algorithms that train an RL agent without a learning signal, i.e., without a task-specific reward function. It encompasses two main research branches: The first is Unsupervised Representation Learning (van den Oord et al., 2019; Ebert et al., 2018; Ha and Schmidhuber, 2018; Laskin et al., 2020; Schwarzer et al., 2021a; Stooke et al., 2021) which focuses on learning a representation of the states or dynamics from noisy or incomplete observations. This line of work includes algorithms that learn world models to train an agent with imagined trajectories to reduce the number of environment interactions (Dockhorn and Apeldoorn, 2018; Hansen et al., 2022; Xu et al., 2022; Rajeswar et al., 2022; Seo et al., 2022). The second branch is called Unsupervised Behavioral Learning.

**Unsupervised Behavioral Learning**  is a variant of the RL problem where the agent can interact with the environment in a reward-free setting before being assigned the downstream task $T$ defined by an external reward function (Oudeyer et al., 2007; Jin et al., 2020). During pretraining, intrinsic rewards are used to train a policy to become an optimal prior and initialization for the subsequent finetuning task. We denote the policy after finetuning as $\pi_T^*$ and the corresponding oracle policy that acts optimally on task $T$ as $\pi_T^+$. Algorithms for Unsupervised Behavioral Learning can be categorized into knowledge-, data- and competence-based approaches (Srinivas and Abbeel, 2021). **Knowledge-based** algorithms define a self-supervised task by making predictions on some aspect of the environment (Pathak et al., 2017; Chen et al., 2022a;b). The approach **data-based** methods follow is to maximize the state visitation entropy to explore

the environment (Zhang et al., 2020; Guo et al., 2021; Mutti et al., 2021; 2022; Hazan et al., 2019; Zhao et al., 2022; Jacq et al., 2022; Nedergaard and Cook, 2022). A recent approach (He et al., 2022) maximizes the Wasserstein distance of the induced state-marginals instead. **Competence-based** algorithms learn skills that maximize the mutual information between the trajectories and a space of skills (Mohamed and Rezende, 2015; Gregor et al., 2017; Baumli et al., 2021; Jiang et al., 2022; Zeng et al., 2022; Shafiullah and Pinto, 2022; Cho et al., 2022). Some algorithms relax the requirement of a task-agnostic prior by selecting the most promising skill before finetuning on the task (RHIM et al., 2022; Laskin et al., 2022). All categories have one thing in common: They traverse the policy space during pretraining and potentially encounter behavior close to the optimal prior policy for the finetuning tasks. There are also algorithms that show great performance by combining both branches of URL by simultaneously learning the dynamics and possible behaviors in a domain (Sharma et al., 2020; Yuan et al., 2022). In this work, however, we focus on model-free approaches.

## 3 Method

In the following, we first discuss the motivation of our approach by linking it to the theoretical results of Eysenbach et al. (2022). Based on that, we derive the regularization strategy of POLTER using a mixture ensemble policy and discuss the concrete implementation of our method. To ground our theoretical motivation, we show exemplary behaviour of URL algorithms with and without POLTER on simple control tasks.

### 3.1 Motivation

The adaptation objective by Eysenbach et al. (2022) connects the policy prior after pretraining $\pi(s)$ with the finetuned policy that has been adapted to an arbitrary downstream task $\pi_T^*(s)$. As the reward function can be identified with a state-occupancy measure and because there are multiple policies that induce the same state-occupancy measure, the adaptation objective is defined in terms of these state-marginals:

$$\min_{\rho_T^*(s)\in\mathcal{C}} \max_{\rho_T^+(s)\in\mathcal{C}} \underbrace{\mathbb{E}_{s\sim\rho_T^+(s)}[r(s)] - \mathbb{E}_{s\sim\rho_T^*(s)}[r(s)]}_{\text{worst-case regret}} + \underbrace{D_{\mathrm{KL}}(\rho_T^*(s) \parallel \rho(s))}_{\text{information cost}} \tag{1}$$

The first term in Equation (1) represents the regret that a finetuned policy $\pi_T^*(s)$ achieves with respect to an oracle policy $\pi_T^+(s)$ for the given task $T$. The second term defines the information cost between $\pi_T^*(s)$ and the prior after pretraining $\pi(s)$ via their induced state-marginals $\rho_T^*(s)$ and $\rho(s)$. The set of all feasible state-marginals in an environment is denoted as $\mathcal{C}$.

Competence-based URL algorithms learn a set of skills $z \in \mathcal{Z}$ during pretraining by maximizing the mutual information $I(\rho(s); z)$ between these skills and the state-marginal.[1] We use the non-standard notation $I(\rho(\cdot); \cdot)$ to differentiate between the mutual information of different state-marginals. For these algorithms, Eysenbach et al. demonstrated that the state-marginal averaged over all skills minimizes the distance to the furthest skill. Note, that this skill is generally unknown and depends on the geometric structure of the state-marginal space of the domain:

**Lemma 1** (Lemma 6.5 in Eysenbach et al. (2022), Theorem 13.1.1 in Cover and Thomas (2006)). *Let $\rho(s \mid z)$ be given. Maximizing the mutual information is equivalent to minimizing the divergence between the average state-marginal $\rho(s)$ and the state-marginal of the furthest possible skill $z \in \mathcal{Z}$:*

$$\max_{p(z)} I(\rho(s); z) = \min_{\rho(s)} \max_z D_{KL}(\rho(s \mid z) \parallel \rho(s)) \tag{2}$$

Thus, the average state-marginal inducing policy minimizes the worst-case information cost of adapting the policy after pretraining to the downstream task $T$. Competence-based algorithms estimate this optimal prior policy by conditioning their policy on the average of the skill space that they have learned during pretraining.[2]

---

[1]In practice, this is achieved by conditioning the policy on a vector that represents the skill.
[2]For example, if the skill space $\mathcal{Z}$ lies within a unit hypercube, the vector in the center of the hypercube is chosen.

Data- and knowledge-based algorithms, however, do not try to estimate the optimal prior and thus perform worse in the URL setting (Laskin et al., 2022). Additionally, these algorithms explore the policy space of the domain on a random trajectory that might be close to the optimal prior at one point, but can deviate from it arbitrarily everywhere else. These disadvantages motivate our key idea to improve the sample-efficiency of data- and knowledge-based URL algorithms.

### 3.2 POLTER: Policy Trajectory Ensemble Regularization

In this section, we adapt the results from Section 3.1 for data- and knowledge-based URL algorithms and derive our regularization POLTER. The mutual information in Lemma 1 can be decomposed in the entropy of the state-marginal and a conditional entropy term:

$$I\left(\rho(s); z\right) = H\left(\rho(s)\right) - H\left(p(\rho(s) \mid z)\right) \tag{3}$$

Data-based algorithms maximize the first term in Equation (3), i.e. they maximize an upper bound of the mutual information by maximizing the entropy of the state-marginal distribution. For knowledge-based algorithms this connection is not as explicit. However, our experiments in Appendix C show that they also achieve a similar state-marginal entropy. Both categories of URL algorithms use a mechanism $\mathcal{M}$ (e.g. the prediction error of a dynamics model) that encodes their objective as an intrinsic reward that the RL agent is trying to maximize during pretraining. To apply Lemma 1 to these algorithms, we have to show that the conditional entropy vanishes.

Our assumptions can be expressed as the latent variable model in Figure 2 where everything is observed except for the skill $z$ which determines the intrinsic return $r_{\text{int}}$. At different points during the pretraining steps $0 \leq t < N_{\text{PT}}$, we observe different intrinsic returns for taking an action $a$ in a state $s$. Thus, the underlying skill that the agent is supposed to learn must have changed. This leads us to the following result:

**Proposition 1.** *Let a mechanism $\mathcal{M}$ be given that provides an intrinsic return. At a fixed point during pretraining, this mechanism defines a reward function $r_i(s)$. Each reward function can be identified with a latent skill $z_i$, i.e. there exists a mapping between the set of reward functions and the set of skills. Let a policy $\pi_i(s)$ with state-marginal $\rho_i(s)$ that maximizes this reward function be given. Then the conditional entropy of the optimal state-marginal and the skill $H\left(p(\rho_i(s) \mid z_i)\right)$ is equal to 0.*

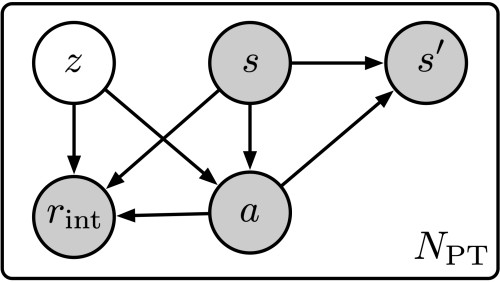

Figure 2: Probabilistic graphical model that underlies the assumption of POLTER for approximating the optimal policy using an ensemble of pretraining policies. The latent skill $z$ determines the intrinsic return $r_{\text{int}}$ via the pretraining objective, and the actions $a$ of the agent's policy are implicitly conditioned on it.

*Proof Sketch.* To maximize the intrinsic return for a given skill $z_i$, the optimal policy $\pi_i(s)$ has to place all of the probability mass on the optimal state-marginal, i.e. $H\left(p(\rho_i(s) \mid z_i)\right) = \mathbb{E}[-\log p(\rho_i(s) \mid z_i)] = \mathbb{E}[-\log 1] = 0$. If there is a policy that induces a different conditional state-marginal distribution with higher return, then the policy $\pi_i(s)$ is not optimal and there is a contradiction. $\qquad\square$

Thus, under Proposition 1 and given an agent that learns the optimal policy for the current state of the intrinsic reward mechanism $\mathcal{M}$, the conditional entropy in Equation (3) is minimized. Given these assumptions, the data- and knowledge-based URL algorithms also maximize the mutual information between the state-marginal and the latent skill and we can apply Lemma 1 to them. This suggests that estimating the policy which induces the average state distribution is possible by selecting policies during pretraining that maximize different reward functions, i.e. learn different latent skills. Therefore, we can approximate the optimal prior policy by an ensemble consisting of a set of policies from different points of pretraining.

**Implementation**    The policy loss $\mathcal{L}^{\mathrm{URL}}$ of the RL agent is augmented by our approach POLTER with the following regularization term:

$$\mathcal{L}^{\mathrm{POLTER}}(\pi) = \mathcal{L}^{\mathrm{URL}}(\pi) + \alpha D_{\mathrm{KL}}(\tilde{\pi} \parallel \pi) \tag{4}$$

where $\tilde{\pi}(s) = \sum_k \phi_k \pi_k(s)$ is the mixture ensemble policy consisting of $k$ previous policies from the URL trajectory. The mixture components $\phi_k = \frac{1}{k}$ parametrize a categorical distribution to construct the non-markovian average ensemble policy. At the start of each episode, the ensemble policy samples the index of an ensemble member with probability $\frac{1}{k}$ and follows this policy afterwards. In the limit, this ensemble policy induces the average state-marginal (see Appendix A.2 in Eysenbach et al. (2022)).

---

**Algorithm 1** URL+POLTER

---

**Require:** Initialize URL algorithm, policy $\pi_{\theta,0}$, replay buffer $\mathcal{D}$, pretraining steps $N_{\mathrm{PT}}$ ⊳ URL init
**Require:** Empty ensemble $E = \emptyset$, ensemble snapshot time steps $\mathcal{T}_E$, regularization weight $\alpha$ ⊳ POLTER init
1: **for** $t = 0 \dots N_{\mathrm{PT}} - 1$ **do** ⊳ Unsupervised pretraining
2:     **if** Beginning of episode **then**
3:         Observe initial state $s_t \sim p_0(s_0)$
4:         **if** $t \in \mathcal{T}_E$ **then** ⊳ Update ensemble policy
5:             Extend ensemble $E \leftarrow E \cup \pi_{\theta,t}$
6:             Update ensemble policy $\tilde{\pi}$
7:     Choose action $a_t \leftarrow \pi_{\theta,t}(a_t \mid s_t)$
8:     Observe next state $s_{t+1} \sim p(s_{t+1} \mid s_t, a_t)$
9:     Add transition to replay buffer $\mathcal{D} \leftarrow \mathcal{D} \cup (s_t, a_t, s_{t+1})$
10:    Sample a minibatch $B \sim \mathcal{D}$
11:    Compute loss $\mathcal{L}^{\mathrm{POLTER}} = \mathcal{L}^{\mathrm{URL}}(\pi_{\theta,t}) + \alpha D_{\mathrm{KL}}(\tilde{\pi} \parallel \pi_{\theta,t})$
12:    Update policy $\pi_{\theta,t}$ with $\mathcal{L}^{\mathrm{POLTER}}$
13: ... ⊳ Supervised finetuning on task T

---

In Figure 1, we illustrate the progression of the optimization trajectory in the induced state-marginal space $\rho(s)$ of a policy $\pi(s)$ under the URL objective with POLTER regularization. Along its trajectory, the policy $\pi(s)$ is attracted to the ensemble policy $\tilde{\pi}$ and, thus, closer to the optimal prior $\pi_T^*$ for a specific finetuning task $T$. This results in a reduced information cost to adapt the pretraining policy to the task and improves the sample-efficiency of the regularized URL algorithm.

We outline the combination of URL with our regularization POLTER in Algorithm 1. We fix the ensemble policy to a finite set of states to stabilize training, i.e., each time we add a member to the ensemble (Line 5), we condition the ensemble policy on the next state. Finally, we compute the Kullback–Leibler divergence (KL-divergence) loss and add it to the policy loss of the URL algorithm (Line 11). The memory overhead is dominated by the requirement to store the ensemble policies. For the computational overhead, we only have to consider the $k$ forward passes for conditioning the ensemble policy on a state when adding a new member.[3]

### 3.3   Demonstration on PointMass and Pendulum

**PointMass**    To ground our theoretical motivation in empirical evidence, we evaluate the effect of our regularization on Random Network Distillation (RND) (Burda et al., 2019), a knowledge-based URL algorithm, and ProtoRL (ProtoRL) (Yarats et al., 2021), a data-based URL algorithm, in the simplistic PointMass environment (Tassa et al., 2018; 2020). In PointMass, we can define the optimal prior $\pi_T^*(s)$ that maximizes the average performance for the downstream finetuning tasks. This optimal prior policy moves the ball into the center of the area, which minimizes the distance to any possible target location that represent a finetuning task. With the optimal prior, we can compute the KL-divergence between the optimal prior policy and the current pretraining policy $D_{\mathrm{KL}}(\pi_T^* \parallel \pi)$. With POLTER, the KL-divergence to the optimal prior policy $\pi_T^*$ is generally lower throughout the whole pretraining phase.

---

[3]In our experiments, each policy takes neglected amount of around 4MB of storage and the wall-clock time increases by 12%.

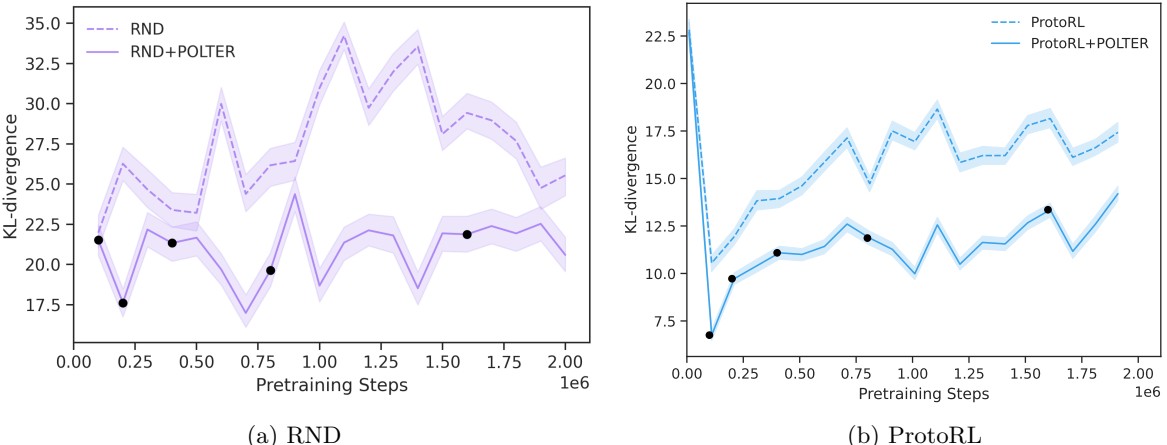

(a) RND

(b) ProtoRL

Figure 3: Average KL-divergence of RND/ProtoRL (dashed) and RND+POLTER/ProtoRL+POLTER (solid) between the pretraining policy $\pi(s_0)$ and the optimal pretraining policy $\pi_T^*(s_0)$ in the PointMass domain during reward-free pretraining. The shaded area indicates the standard error over 10 seeds and the dots are the pretraining checkpoints that were used for the ensemble.

**Pendulum** As described in Figure 1, the effect of POLTER on the performance on a specific finetuning task in a given domain depends on the optimal policy for the given task. Thus, even though POLTER improves the average performance over all possible tasks, it might have no or even a detrimental effect on single downstream tasks. To test this hypothesis, we train POLTER on the Pendulum domain for two different downstream tasks. The first one is the known *SwingUp* task, where the agent has to balance the pole in an upright position. The second task is a *Propeller* task, which rewards the maximization of the angular velocity of the pole. This task's optimal policy induces a state-marginal that is further away from the average, which is reflected in Figure 4 in a slightly decreased performance when applying POLTER to RND. However, POLTER consistently increases the performance on the regular *SwingUp* task. Further results for Contrastive Intrinsic Control (CIC) (Laskin et al., 2022) and ProtoRL (Yarats et al., 2021) in this experiment can be found in Appendix B.

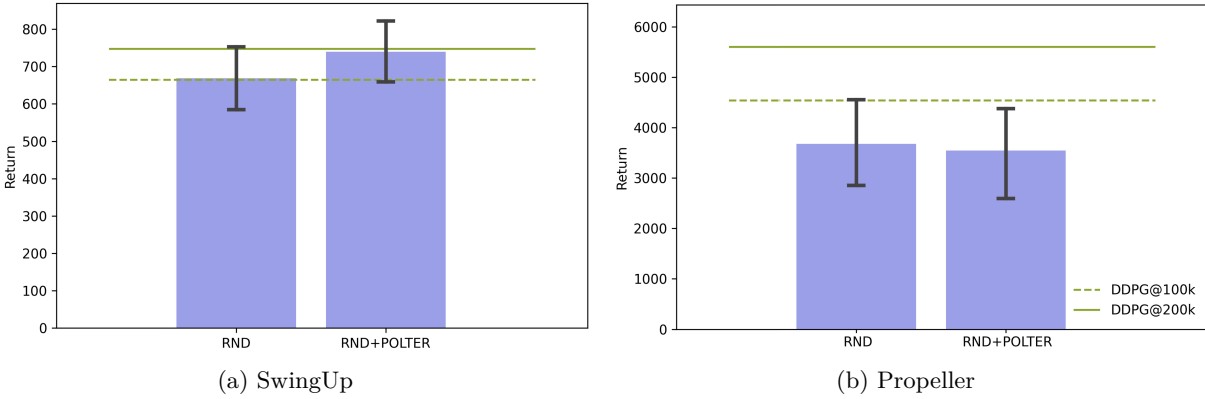

(a) SwingUp

(b) Propeller

Figure 4: Finetuning performance of RND and RND+POLTER in the Pendulum domain averaged over 10 seeds after 100 k of pretraining. The baseline of Deep Deterministic Policy Gradient (DDPG) is trained for 100 k or 200 k steps without pretraining. The error bars indicate the standard error over 10 seeds.

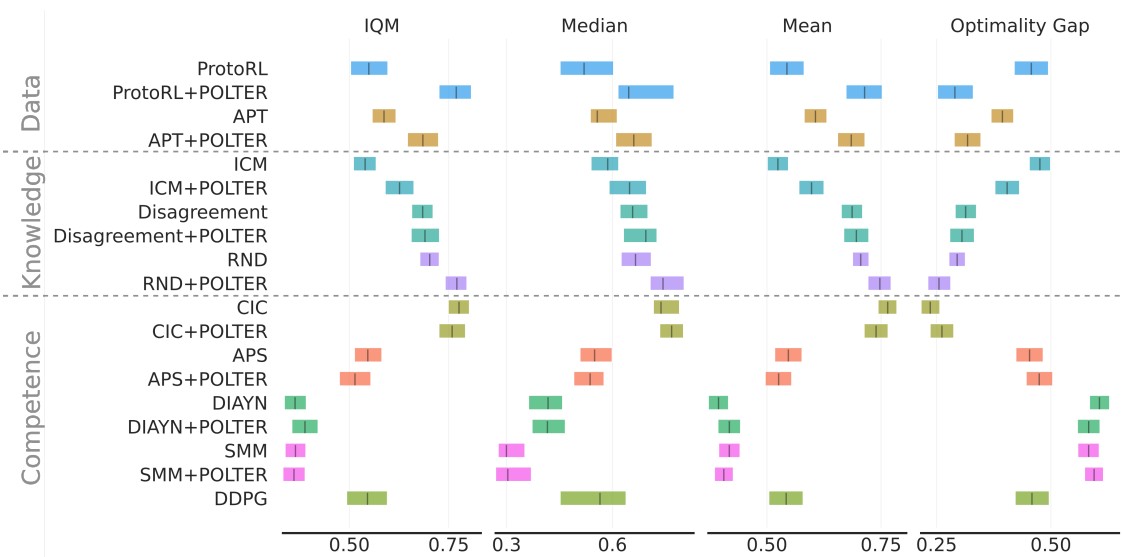

Figure 5: Aggregate statistics of applying POLTER to several URL algorithms after pretraining for 2 M steps. POLTER improves the performance of most algorithms substantially, for ProtoRL even by 40%. Each algorithm is tested on the URLB with its 12 tasks from 3 domains using 10 independent seeds, resulting in 120 runs per algorithm. The error bars indicate the 95% bootstrap confidence intervals. DDPG is the baseline without pretraining.

## 4 Experiments

With our theoretical motivation and our prospect on the toy environments PointMass and Pendulum, we now address the following empirical questions at a larger scale: **(Q1)** How does POLTER affect the performance after pretraining, and does the performance differ across URL algorithm categories? **(Q2)** Does POLTER improve the sample-efficiency for finetuning? **(Q3)** How does the performance vary with an increasing number of pretraining steps? **(Q4)** How does the strength of the regularization affect the performance of different algorithms in different domains? **(Q5)** How to select the checkpoints for the ensemble policy in POLTER? **(Q6)** Do the results also hold for pixel-based observations?

To answer these questions, we evaluate several different URL algorithms with and without our POLTER regularization on the Unsupervised Reinforcement Learning Benchmark (Laskin et al., 2021). We use the provided code from Laskin et al. (2021) to aid reproducibility and provide our source code in the supplementary material. Knowledge-based algorithms, such as RND (Burda et al., 2019), Disagreement (Pathak et al., 2019), and Intrinsic Curiosity Module (ICM) (Pathak et al., 2017), use the norm or variance of their prediction errors of some aspect of the environment as a learning signal. The second category consisting of Active Pre-training (APT) (Schwarzer et al., 2021b), and ProtoRL (Yarats et al., 2021), are data-based algorithms, that try to maximize the entropy of the state-visitation distribution to explore the environment. The last category is competence-based algorithms that try to learn a set of skills by maximizing the mutual information between a learned skill vector and some features of the observed states. Besides the three algorithms that were used by Laskin et al. (2021), Active Pre-training with Successor Features (APS) (Liu and Abbeel, 2021), State Marginal Matching (SMM) (Lee et al., 2020), and "Diversity is All You Need" (DIAYN) (Eysenbach et al., 2019), we evaluate our method with a newer method from this category, called CIC (Laskin et al., 2022).

## 5 Results

**Evaluation** For evaluation, the agent performs pretraining in a reward-free setting for a total of $N_{PT} = 2$ M steps in one of the three domains using one of the described algorithms with or without our POLTER

regularization. The performance is then measured as the average return over 10 episodes after finetuning for 100 k steps given external rewards that describe the downstream task. We follow the exact evaluation protocol of the URLB and use DDPG (Lillicrap et al., 2016) on state-based observations. For its hyperparameters and the setup of POLTER, see Appendix E. We follow Agarwal et al. (2021) in their evaluation using the IQM as our main metric.[4] The expert performance is provided by Laskin et al. (2021). Each algorithm is tested on 12 tasks from 3 domains using 10 independent seeds, resulting in 120 runs per algorithm.

### Q1: How does POLTER affect the performance of different algorithm categories?

We report the finetuning performance of each URL algorithm after 2 M pretraining steps with and without POLTER in Figure 5. With our method, we see an increase in performance for data- and knowledge-based algorithms by 19% IQM on average. For the data-based method ProtoRL pretraining with POLTER increases the performance even by 0.22 IQM (40%). This result supports the argument in Section 3.2, as both data-based algorithms directly maximize the entropy of the state-marginal distribution and with it the upper bound of the mutual information between the state-marginal and the latent skill. The improvement due to POLTER also depends on the exploration capabilities of the URL algorithm. This can be seen for Disagreement+POLTER, which performs worst of all knowledge-based algorithms and is due to the limited exploration of the state space, as has recently been shown by Yarats et al. (2022). As noted by Laskin et al. (2021), the baseline performances notably differ among themselves, and most competence-based methods are worse than the supervised DDPG baseline without pretraining. The application of POLTER has no noticeable effect on their overall performance. This is due to the similar effect of their skill-averaging and the ensemble, which both pull the policy to the optimal prior. However, in Section 5, we show that POLTER can be beneficial even for these algorithms. For more detailed results see Appendix F and Section 5.

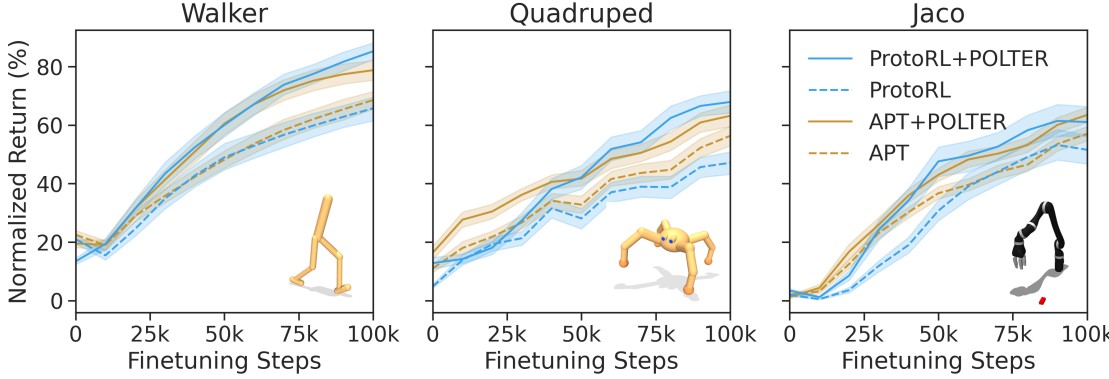

Figure 6: Data-based URL algorithms with (solid) and without POLTER (dashed) regularization after 2 M steps of pretraining. The shaded area indicates the standard error. POLTER speeds up finetuning by ≈ 40% on Walker and Quadruped and ≈ 10% on Jaco.

### Q2: Does POLTER improve the sample-efficiency for finetuning?

As described in Section 3, we try to reduce the information cost of finetuning a prior policy on a specific task to improve the sample-efficiency. Because the effect of our method is most pronounced in the data-based algorithm category, in Figure 6 we show the expert normalized return of finetuning APT and ProtoRL with and without POLTER after 2 M steps of pretraining. The improvement in sample-efficiency is clearly visible, especially in the locomotion domains. Here, POLTER achieves a speed-up of ≈ 40% compared to the unregularized URL algorithms. It should be noted that the initial performance directly after pretraining is similar for all variants, and they all begin to improve after the same number of finetuning steps. This implies that exploration during finetuning is not the deciding factor for the performance differences.

---

[4]Although we used the provided code by Laskin et al. (2021; 2022) for all baselines, our reproduced results slightly differ from their reported performance.

**Q3: How does the performance vary with an increasing number of pretraining steps?**

In their original publication, Laskin et al. (2021) noted that the performance of most URL algorithms decreased with an increasing number of pretraining steps. To evaluate the effect of applying POLTER during pretraining, we observe the performance at different numbers of pretraining frames for the three domains of URLB. In Figure 7, we present our results for knowledge- and data-based methods. Applying POLTER (solid) improves performance at all steps over the baseline (dashed) during pretraining, especially in the locomotion domains. Jaco is a special case because the arm is in a fixed location, and the return is dominated by the ability to bring the arm into a specific position. Thus, the domain relies much less on deeper exploration compared to the locomotion environments, and the initial amount of pretraining for 100 k steps is sufficient to improve on the DDPG baseline without pretraining.

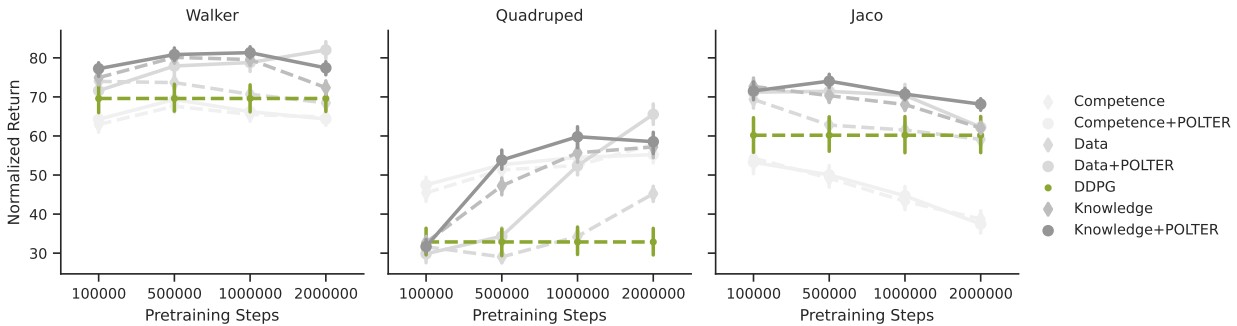

Figure 7: Normalized return per task and algorithm category with (solid) and without (dashed) POLTER regularization after finetuning from different pretraining snapshots. DDPG is the baseline without pretraining.

The results in Figure 7 also show an increase of the effect of POLTER on the downstream performance with an increasing number of pretraining steps. This leads to the question of what specific effect POLTER has on the URL algorithm and in particular on the actor network where the regularisation term is applied. In Figure 8, we computed the variance of the actor gradients over a window of 5 gradient steps for RND and ProtoRL with and without POLTER. The results support the findings of Laskin et al. that longer pretraining is often not beneficial for the downstream performance, as the variance of the gradients and thus, the changes to the network weights decrease substantially for both baselines. With POLTER, however, the variance of the gradients does not decrease as much. It is also apparent that the difference between POLTER and the baseline is much more pronounced for ProtoRL than for RND. This result is also in line with the detailed analysis in Figure 18.

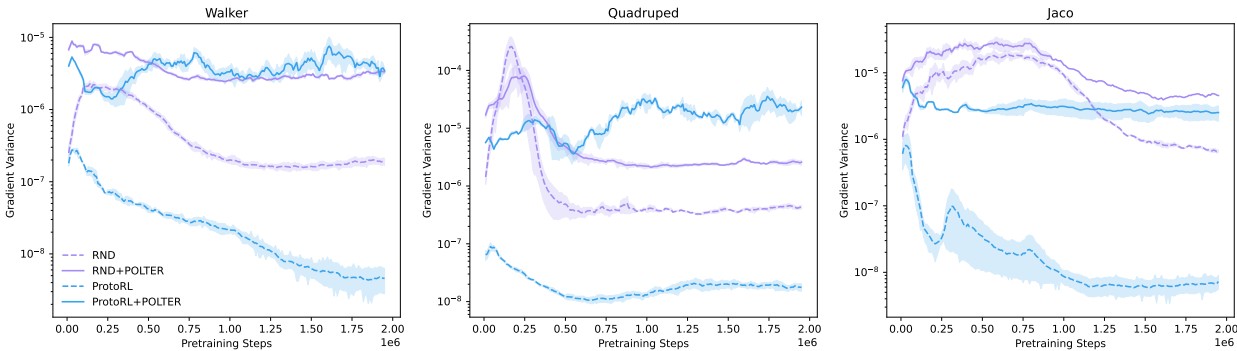

Figure 8: Variance of the actor gradients for ProtoRL and RND with and without POLTER regularization on the URLB averaged over 10 seeds with a sliding window size of 5. The shaded area indicates the standard error of the mean.

**Q4: How does the strength of the regularization affect the performance?**

A hyperparameter of our method is the strength $\alpha$ of our regularization, see Equation (4). We set this to 1.0 in our preceding evaluations as a natural choice. To study the sensitivity of this hyperparameter, we evaluated POLTER with different settings on a set of representative URL algorithms. In Figure 9, we present

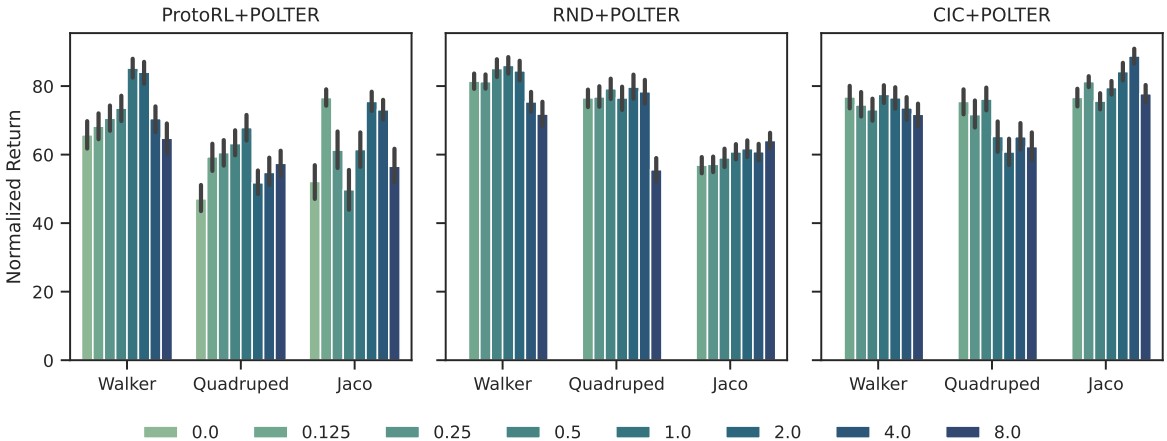

Figure 9: Average normalized return of ProtoRL+POLTER, RND+POLTER and CIC+POLTER after 2 M steps of pretraining and 100 k finetuning for different values of the regularization strength $\alpha$.

the performance of the POLTER regularized variants of ProtoRL, RND and CIC over different values of $\alpha$. The results show a robust performance of POLTER with its default $\alpha = 1.0$, i.e., POLTER performs better or is on par compared to not using POLTER ($\alpha = 0.0$). On average, $\alpha = 1.0$ seems to be the best choice. Nevertheless, the performance of our method can be further improved if it is tuned depending on the task and algorithm. In the Jaco domain, a higher regularization helps RND but might be detrimental for ProtoRL. We hypothesize that this is due to increasingly extreme pretraining policies of RND that are dampened via POLTER, which is helpful for the tasks in this domain. For CIC, the regularization of POLTER has a minor effect or even degrades the performance in the locomotion domains. As Laskin et al. (2022) noted, locomotion requires a high-entropy pretraining policy that explores the environment well. POLTER is indifferent to the skill vector used by the CIC policy in each episode. So the ensemble contains different skill vectors that are likely to suggest different actions, which partly cancel each other and reduce the exploration.

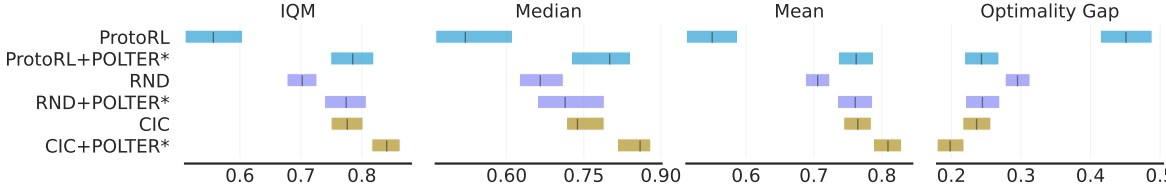

Figure 10: Aggregate statistics of three representative algorithms from each URL category after 2 M steps of pretraining and 100 k finetuning with tuned regularization strength $\alpha$ in Equation (4).

However, a stronger regularization up to a certain value significantly improves the performance on Jaco, as it does not require much exploration to perform well. This demonstrates the trade-off between a good prior for finetuning and an attenuated exploration if the regularization is too strong. It also shows that POLTER should be tuned to the domain.

If we apply the insights from Figure 9, we can improve the performance of the three evaluated algorithms even further by tuning $\alpha$ to the locomotion and manipulation domains.[5] For the results in Figure 10,

---

[5]Tuning here means selecting the best performing $\alpha$ from the previously evaluated grid sweep.

we set $\alpha$ for ProtoRL to 1.0/2.0 (locomotion/manipulation), for RND to 2.0/8.0 and for CIC to 0.0/4.0. We call the tuned variants ProtoRL+POLTER*, RND+POLTER* and CIC+POLTER*. Note that our regularization complements the discriminator loss of CIC, which is set to 0.9/0.0 on the two domain categories. These optimized settings lead to a new state-of-the-art in the URLB of 0.84 IQM. We note that previous state-of-the-art methods such as CIC are also tuned based on the domain of the task and that this experiment allows for a fair comparison.

**Q5: How to select the checkpoints for the ensemble policy in POLTER?**

With POLTER, we introduce the hyperparameter $\mathcal{T}_E$ that determines which checkpoints should be added to the ensemble. As described in Appendix E, we use the same setting for all our experiments by aligning this hyperparameter with the sign-changes of the second derivative in the intrinsic return during pretraining of RND in the Quadruped domain. Following our derivation in Section 3.2, this indicates a change in the latent skill because the actions that the agent takes are now being rated differently by the intrinsic reward mechanism.

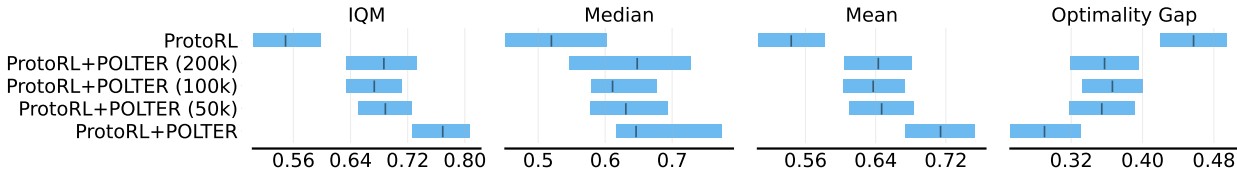

Figure 11: Aggregate statistics on the URLB of different linear and logarithmic checkpoint schedules when applying POLTER to ProtoRL.

To analyze the sensitivity of our regularization to this hyperparameter, in Figure 11 we evaluated ProtoRL on the URLB with three different ensemble schedules. Each schedule adds a checkpoint every $\{50\,\text{k}, 100\,\text{k}, 200\,\text{k}\}$ steps. The results in Figure 11 indicate that POLTER consistently improves the performance of ProtoRL for several settings of this hyperparameter. However, using a logarithmic schedule that aligns with the intrinsic reward dynamics to select the checkpoints further boosts the performance impact of the regularization. Thus, a better understanding of the latent skill space of the domain and algorithm would enable more sophisticated schedules and increase the performance even further.

**Q6: Do the results also hold for pixel-based observations?**

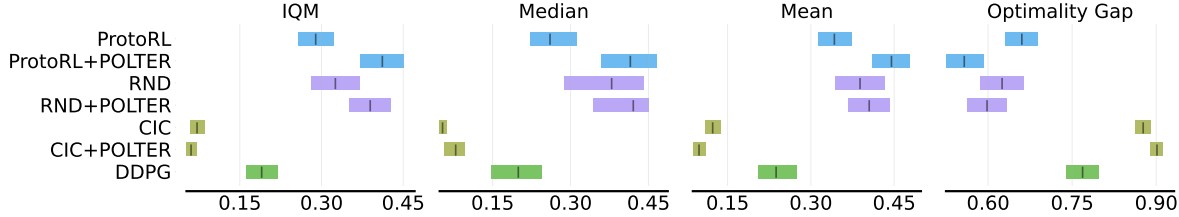

Figure 12: Aggregate statistics of applying POLTER to several URL algorithms after pretraining for $2\,\text{M}$ steps. Each algorithm is tested on the URLB with its 12 tasks from 3 domains using 10 independent seeds with **pixel-based observations**, resulting in 120 runs per algorithm. The error bars indicate the 95% bootstrap confidence intervals. DDPG is the baseline without pretraining.

Laskin et al. (2021) showed that the performance of pixel- and state-based URL algorithms differs substantially. To demonstrate the generality of our approach, we evaluate the effect of POLTER on ProtoRL, RND and CIC on the URLB with pixel-based observations. The results in Figure 12 show that the improvements in IQM are approximately the same for RND and ProtoRL as in state-based environments. For CIC we also

see the same effect of a slightly reduced performance. Note, however, that CIC does not work well for the tasks in the Jaco domain.[6]

# 6 Limitations and Future Work

In this work, we focused our experiments on DDPG to compare our performance on the URLB. This, however, limits our scope to continuous control problems. Therefore, extending URL to environments with discrete action spaces is a possible next step. Moreover, a deeper theoretical understanding of the pretraining processes and the connection to the optimal prior policy would enable more sophisticated mixture policies and a better approximation of the optimal prior in the future. As noted in Section 3, the space of policies is well explored by the URL algorithm, using the average policy is optional under Proposition 1. However, a future research direction is the analysis of the skills that the agent learns during pretraining and incorporate this knowledge into the mixture prior.[7] Finally, POLTER is not adapted to the category of the URL algorithm it is applied to. So its performance with competence-based URL algorithms leaves room for improvement by taking the observed explicit skills into account.

# 7 Conclusion

In this work, we introduced POLTER (Policy Trajectory Ensemble Regularization) – a general method to improve the performance of URL algorithms. URL is a method to increase the sample-efficiency of RL algorithms on a set of tasks by pretraining the policy with an intrinsic reward in the task's domain. Our regularization pulls the pretraining policy closer to an ensemble of pretraining policies seen during pretraining that correspond to a set of explicit or implicit skills. We demonstrated the effect of our method in the PointMass and Pendulum domain and extensively evaluated baseline algorithms regularized with POLTER on the URLB and established its effectiveness in our main experiments. For data- and knowledge-based URL methods, we improved performance on average by 19% and up to 40% (IQM). We argue that our method is a suitable regularization for those algorithms, in contrast to competence-based methods where the effect of POLTER is highly algorithm-dependent. Finally, we showed that our regularization strength can be tuned to the domain and algorithm at hand for further performance improvements, achieving a new state-of-the-art performance with CIC+POLTER*. In future work, this manual tuning could be automated utilizing AutoRL (Parker-Holder et al., 2022). With POLTER's easy implementation and negligible computational requirements, we hope it finds its way into more URL algorithms and spurs further research on how we can learn general priors for arbitrary tasks.

## Acknowledgements

Carolin Benjamins and Marius Lindauer acknowledge funding by the German Research Foundation (DFG) under LI 2801/4-1. Additionally, this work was supported by the Federal Ministry of Education and Research (BMBF), Germany under the project LeibnizKILabor (grant no. 01DD20003) and the AI service center KISSKI (grant no. 01IS22093C), the Center for Digital Innovations (ZDIN) and the Deutsche Forschungsgemeinschaft (DFG) under Germany's Excellence Strategy within the Cluster of Excellence PhoenixD (EXC 2122).

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

# A    Extended Demonstration on PointMass

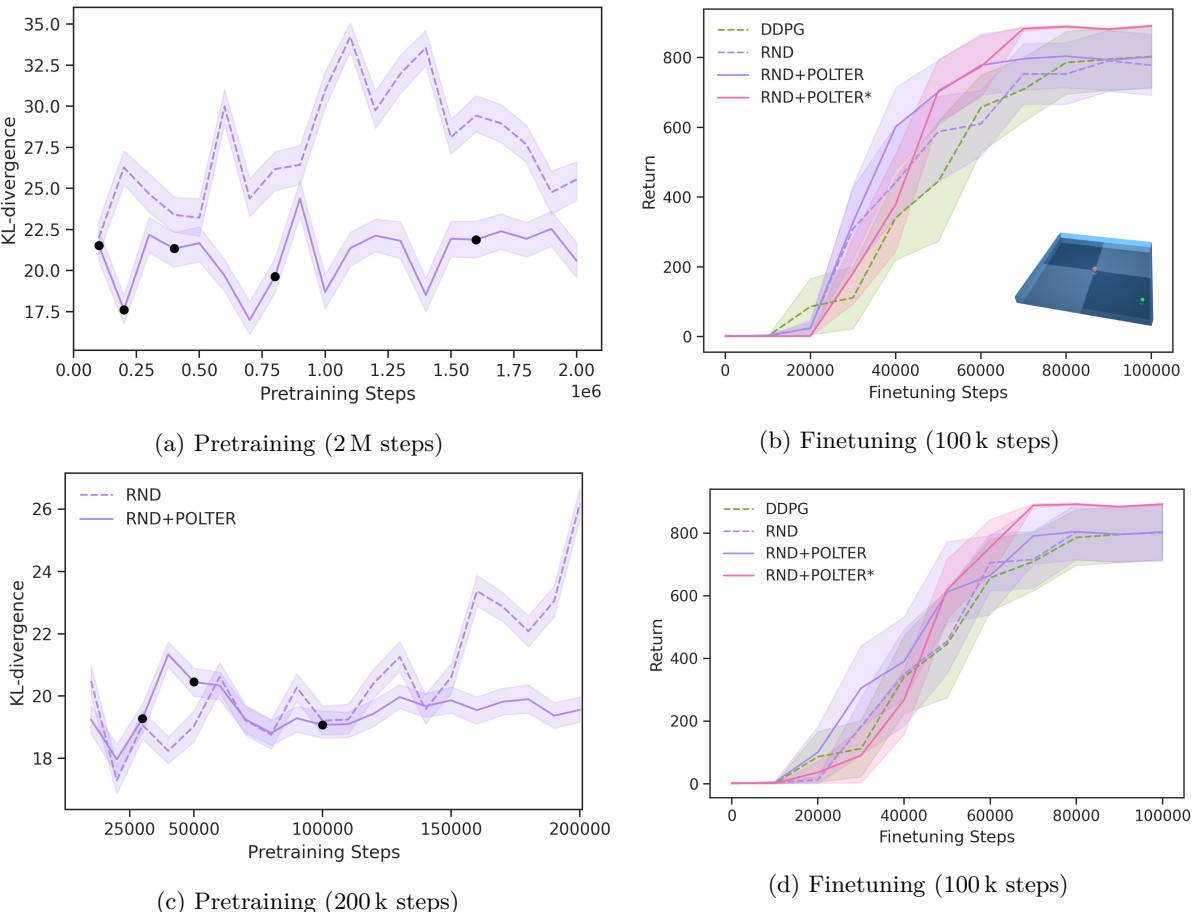

(a) Pretraining (2 M steps)

(b) Finetuning (100 k steps)

(c) Pretraining (200 k steps)

(d) Finetuning (100 k steps)

Figure 13: (**a**), (**c**) Average KL-divergence of RND (dashed) and RND+POLTER (solid) between the pretraining policy $\pi(s_0)$ and the optimal pretraining policy $\pi_T^*(s_0)$ in the PointMass domain during reward-free pretraining. (**a**) is trained for 2 M steps, and (**c**) is trained for 200 k. Each policy is evaluated every 100 k steps on 20 initial states over 10 seeds. The black dots indicate the steps a snapshot is added to the ensemble. (**b**), (**d**) Return during finetuning after 2 M and 200 k pretraining steps, where the target is placed at a fixed random position for each of the 10 seeds. We also provide a DDPG baseline without pretraining and RND+POLTER* using the optimal policy instead of the ensemble. The shaded area indicates the standard error.

We evaluate the effect of our regularization on RND (Burda et al., 2019), a well-known knowledge-based URL algorithm, in the simplistic PointMass environment (Tassa et al., 2018; 2020). In this environment, the agent has to apply a force to a mass (green in Figure 13b) in a 2D plane to reach a target (red) and observes its position and speed. We pretrain a DDPG agent using RND with and without POLTER for 2 M steps and evaluate the policies every 100 k steps, in accordance with the URL benchmark evaluation protocol. Because PointMass is a much simpler environment than the ones contained in URLB and to gain a better resolution for the start of the pretraining, we repeat the experiment with 200 k pretraining and 100 k finetuning steps. We use 10 seeds.

In our whitebox benchmark PointMass, we can train the optimal prior policy and compute the KL-divergence between the optimal prior policy and the current pretraining policy $D_{\mathrm{KL}}(\pi_T^* \parallel \pi)$. Comparing the KL-divergence during pretraining with and without POLTER in Figure 13a shows that with our method, the KL-divergence to the optimal prior policy $\pi_T^*$ is generally lower, which is also apparent for the short pretraining (Figure 13c). POLTER keeps the KL-divergence at bay, whereas RND without POLTER diverges

much more, which is consistent with prior work showing that URL tends to diverge with too many pretraining steps.

## A.1 Effect of POLTER on Finetuning

The effect of this improved prior is also apparent during finetuning in Figure 13b. Here, we see that using an ensemble is a good proxy for the optimal prior due to the similar sample-efficiency. In addition, we observe a speed-up of approximately 25 % (2 M pretraining steps) and 12.5 % (200 k pretraining steps) when using RND+POLTER in reaching the final performance of DDPG.

## A.2 Visualization of State Distributions

In Figure 14a, we can see the state distribution changes during pretraining with and without POLTER. With POLTER, the state space coverage is less, and the trajectories seem more ordered. RND without POLTER also seems to visit the edges often at the end. When using the POLTER regularization, we can see that each pretraining checkpoint is visiting different states, as indicated by the visibility of the previous checkpoint's state visitations. When not using POLTER, we can see that the visitations overlap. Figures 14b and 14c show the discretized position and speed the agent explores throughout pretraining. Especially the discretized speed (Figure 14c) demonstrates the tradeoff between dampening the exploration with POLTER and finding better prior policies because with POLTER, fewer states are frequented, and the states are less extreme.

## B Extended Experiments on Pendulum

In this experiment, we shed further light on the properties of POLTER and its effect under different finetuning tasks. For example, in the PendulumPropeller task, the goal is to maximize the angular velocity of the pole. However, this requires a rather extreme policy, whereas, in the PendulumSwingUp task, the agent has to balance the pole and keep it upright.

The SwingUp task demonstrates that POLTER can consistently improve the performance of the baseline URL agent. However, for the Propeller task, the results are mixed. For ProtoRL, POLTER again shows a great improvement over the baseline. This supports the performance that we observed in the main experiments. But for CIC and RND, POLTER decreases performance. The extreme policy of achieving a high angular velocity is further away from the average, so applying POLTER has a detrimental effect. Nevertheless, note that regardless of this failure case, on average, the tasks will lie closer to the average state distribution, and thus, POLTER will increase the performance.

## C State-Visitation Entropy

To gain further insights into POLTER, we conduct an experiment following Hazan et al. (2019). We discretize the state space of the Walker environment and compute the state-visitation entropy during reward-free pretraining. In Table 1, we see that the entropy of the distribution of POLTER regularized algorithms is lower than that of their counterpart. This effect is most pronounced in data-based algorithms, such as ProtoRL and APT, where the performance is also improved the most (see Table 3).

Table 1: State-visitation entropy of the evaluated URL algorithm categories in the Walker domain during pretraining. Averaged over 10 seeds with 50 k states each at pretraining steps 100 k, 500 k, 1 M and 2 M.

| POLTER | Data | Knowledge | Competence |
|:---:|:---:|:---:|:---:|
| ✘ | $0.2772 \pm 0.0188$ | $0.2863 \pm 0.0036$ | $0.2540 \pm 0.0429$ |
| ✔ | $0.2545 \pm 0.0493$ | $0.2848 \pm 0.0054$ | $0.2511 \pm 0.0422$ |

Knowledge-based algorithms also benefit from the regularization but have a slightly reduced entropy. Because competence-based algorithms already average over a set of skills found during pretraining, the effect of POLTER is the smallest.

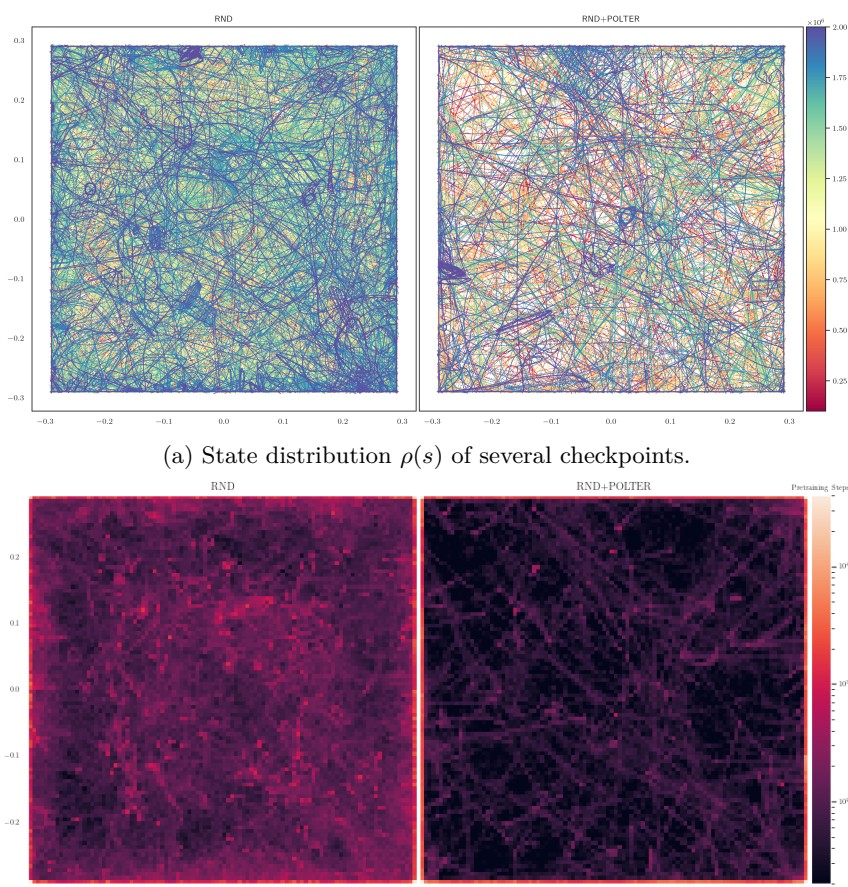

(a) State distribution $\rho(s)$ of several checkpoints.

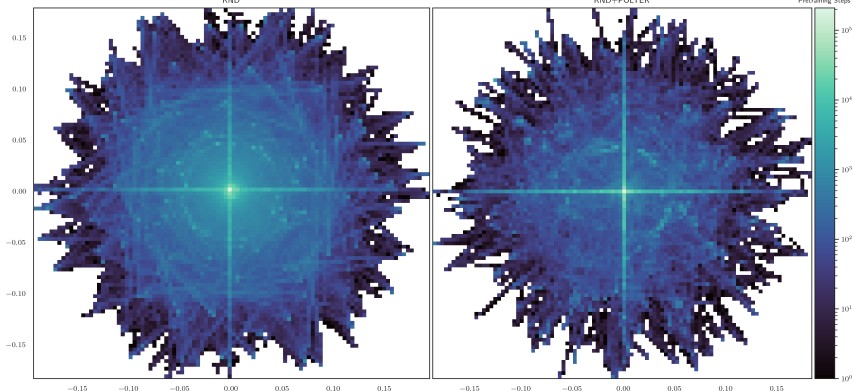

(b) Discretized state histogram of the positions. Looking closely, we can see that the edges of the 2D plane are very often visited. This is due to policies constantly applying the same force and thus reaching the edge of the 2D plane.

(c) Discretized state histogram of the speeds. Note that the prominent horizontal and vertical lines result from moving at the edges and being stuck at the corners of the 2D plane.

Figure 14: State distribution and histogram of RND (always left column) and RND+POLTER (right column) during pretraining on the PointMass environment.

These results imply that POLTER does not lead to a better state-space exploration. Instead, its performance gains are the result of an improved prior as indicated by the reduced KL-divergence between the policy and

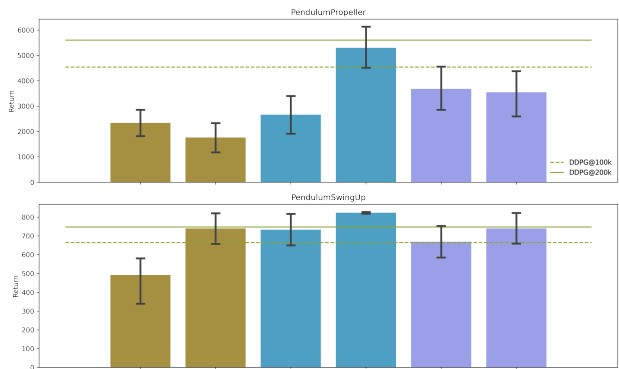

Figure 15: Average return over 10 seeds on two tasks in the Pendulum domain with different characteristics. Algorithms from all three URL categories are compared to a DDPG baseline that is trained for either 100 k or 200 k steps without pretraining.

the optimal pretraining policy on PointMass (Section 3.3). This experiment showed that a good exploration of the state-space is required but not sufficient to achieve good performance with URL algorithms.

## D    Environments in the Unsupervised Reinforcement Learning Benchmark

The Unsupervised Reinforcement Learning Benchmark Laskin et al. (2021) contains three domains with the topics of locomotion and manipulation. **Walker**, **Quadruped** and **Jaco**, are used to explore the effects of different URL algorithms. It has a specific training and evaluation protocol which we also follow in this work. The **Walker** domain contains a planar walker constrained to a 2D vertical plane, with an 18-dimensional observation space and a 6-dimensional action space with three actuators for each leg. The associated tasks are *stand*, *walk*, *run* and *flip*. The walker domain provides a challenging start for the agent since it needs to learn balancing and locomotion skills to be able to adapt to the given tasks. The next domain is **Quadruped**, which expands to the 3D space. It has a much larger state space of 56 dimensions and a 12-dimensional action space with three actuators for each leg. The tasks in this environment are *stand*, *walk*, *jump* and *run*. The last environment used is the **Jaco Arm**, which is a robotic arm with 6-DOF and a three-finger gripper. This domain is very different from the other two, as its setting is manipulation and not locomotion. The tasks are *Reach top left*, *Reach top right*, *Reach bottom left* and *Reach bottom right*.

## E    Hyperparameters and Resources

**POLTER Hyperparameters**    During pretraining, we construct the mixture ensemble policy $\tilde{\pi}$ with $k = 7$ members at specific time steps $\mathcal{T}_E$. For adding each member we choose the ensemble snapshot time steps $\mathcal{T}_E = \{25\,\mathrm{k}, 50\,\mathrm{k}, 100\,\mathrm{k}, 200\,\mathrm{k}, 400\,\mathrm{k}, 800\,\mathrm{k}, 1.6\,\mathrm{M}\}$. The steps were chosen according to initial experiments of applying RND in the Quadruped domain, where there are large changes of the intrinsic reward at the beginning, which become progressively smaller over time. We set the regularization strength $\alpha = 1$ and use the same hyperparameters for each of the three domains unless specified otherwise.

**Baseline Hyperparameters**    The hyperparameters for our baseline algorithms follow Laskin et al. (2021) and Laskin et al. (2022). The hyperparameters for the DDPG baseline agent are described in Table 2.

**Compute Resources**    All experiments were run on our internal compute cluster on NVIDIA RTX 1080 Ti and NVIDIA RTX 2080 Ti GPUs and had 64GB of RAM and 10 CPU cores. In total, we trained over $12\,000$ models and performed $\approx 3\,500\,200\,000$ environment steps.

Table 2: Hyperparameters for the DDPG algorithm.

| Hyperparameter | Value |
| --- | --- |
| Replay buffer capacity | $1 \times 10^6$ |
| Action repeat | 1 |
| Seed frames | 4000 |
| $n$-step returns | 3 |
| Batch size | 1024 |
| Discount factor $\gamma$ | 0.99 |
| Optimizer | Adam |
| Learning rate | $1 \times 10^{-4}$ |
| Agent update frequency | 2 |
| Critic target EMA rate | 0.01 |
| Feature size | 1024 |
| Hidden size | 1024 |
| Exploration noise std clip | 0.3 |
| Exploration noise std value | 0.2 |
| Pretraining frames | $2 \times 10^6$ |
| Finetuning frames | $1 \times 10^5$ |

## F  Detailed Results on Unsupervised Reinforcement Learning Benchmark

This section provides additional results for our experiments on Unsupervised Reinforcement Learning Benchmark. In the supplementary, we provide the raw scores. The statistics comparing URL algorithms with and without POLTER aggregated for finetuning on 12 tasks across 10 seeds can be found in Table 3. In addition we show aggregate statistics of the absolute improvement in expert performance in Figure 16 and the performance profiles (Agarwal et al., 2021) per URL algorithm category in Figure 17. As before, we see a large improvement for data- and knowledge-based algorithms and a small or negative for competence-based algorithms. The improvement sometimes varies strongly across seeds and tasks. Also, we show the normalized return after finetuning from different pretraining snapshots for each domain and URL category in Figure 18. In the Jaco domain, URL algorithms with and without POLTER mostly deteriorate with an increasing number of pretraining steps. Each category shows a different trend in each domain. Interestingly, the competence-based algorithms SMM and DIAYN fail during pretraining in the Jaco domain. In Figure 19 we see the normalized return over finetuning steps. POLTER is mostly on par or speeds up compared to the URL algorithm without POLTER. In total, most algorithms do not converge yet after 100 k steps.

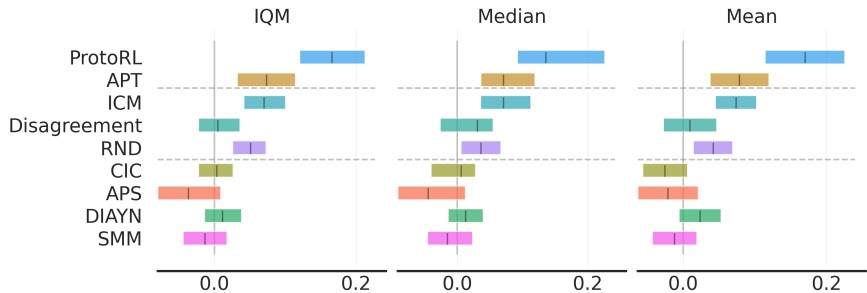

Figure 16: Aggregate statistics of the absolute improvement with POLTER per URL category.

Table 3: Raw aggregate statistics following Agarwal et al. (2021) of evaluated URL algorithms with and without POLTER regularization. The results marked with POLTER* were obtained by tuning the regularization strength to the task domain of locomotion (Walker and Quadruped) and manipulation (Jaco).

| | IQM ↑ | Mean ↑ | Median ↑ | Optimality Gap ↓ | POLTER IQM Improvement |
|---|---|---|---|---|---|
| **Algorithm** | | | | | |
| ProtoRL | 0.56 | 0.55 | 0.52 | 0.45 | |
| ProtoRL+POLTER | 0.77 | 0.71 | 0.65 | 0.29 | +40% |
| ProtoRL+POLTER* | 0.79 | 0.76 | 0.80 | 0.24 | +41% |
| APT | 0.59 | 0.61 | 0.56 | 0.39 | |
| APT+POLTER | 0.69 | 0.68 | 0.66 | 0.32 | +17% |
| RND | 0.70 | 0.71 | 0.67 | 0.30 | |
| RND+POLTER | 0.77 | 0.75 | 0.74 | 0.26 | +10% |
| RND+POLTER* | 0.77 | 0.76 | 0.71 | 0.24 | +10% |
| ICM | 0.54 | 0.52 | 0.59 | 0.48 | |
| ICM+POLTER | 0.63 | 0.60 | 0.65 | 0.40 | +17% |
| Disagreement | 0.68 | 0.69 | 0.66 | 0.31 | |
| Disagreement+POLTER | 0.69 | 0.70 | 0.69 | 0.31 | +1% |
| CIC | 0.78 | 0.76 | 0.74 | 0.24 | |
| CIC+POLTER | 0.76 | 0.74 | 0.77 | 0.26 | -2% |
| CIC+POLTER* | 0.84 | 0.81 | 0.86 | 0.20 | +7% |
| DIAYN | 0.36 | 0.39 | 0.42 | 0.61 | |
| DIAYN+POLTER | 0.39 | 0.42 | 0.42 | 0.58 | +8% |
| SMM | 0.36 | 0.42 | 0.30 | 0.58 | |
| SMM+POLTER | 0.36 | 0.41 | 0.30 | 0.59 | ± 0% |
| APS | 0.56 | 0.58 | 0.55 | 0.42 | |
| APS+POLTER | 0.52 | 0.53 | 0.54 | 0.47 | -7% |
| DDPG | 0.55 | 0.54 | 0.56 | 0.46 | |

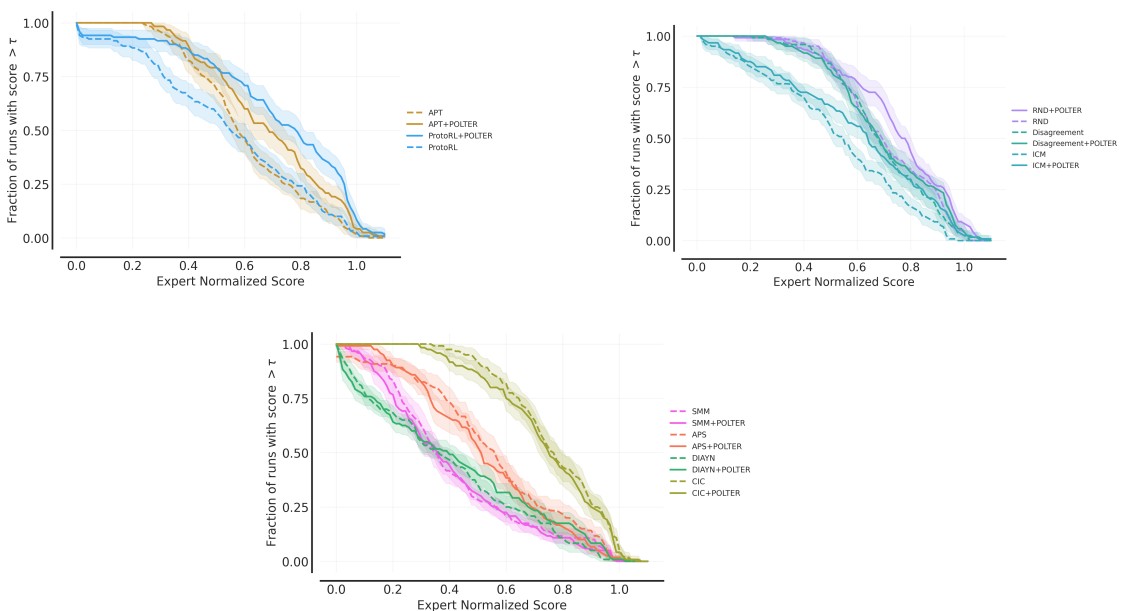

Figure 17: Performance profiles after finetuning of the different algorithms averaged over 10 seeds where the shaded region indicates the standard error. Variants without POLTER are dashed.

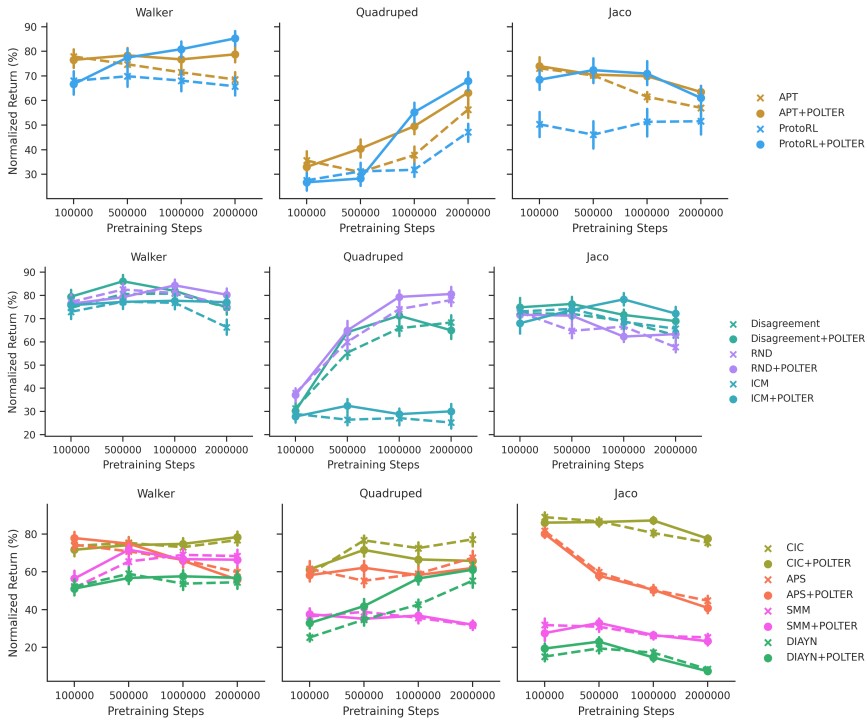

Figure 18: Finetuning from different pretraining snapshots of data-, knowledge- and competence-based algorithms. The error bars indicate the standard error of the mean.

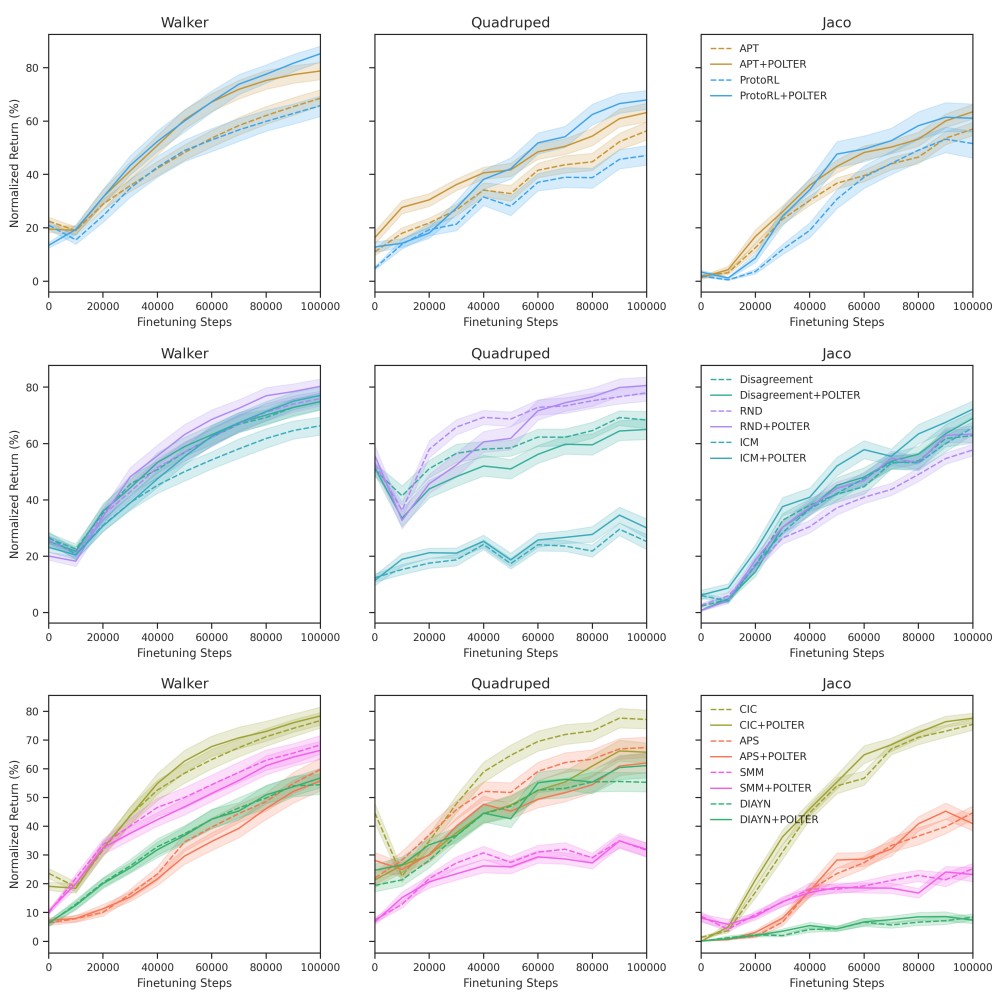

Figure 19: Finetuning curves of data-, knowledge- and competence-based algorithms after pretraining for 2 M steps. The shaded area indicates the standard error.

