# OpenReview forum: "POLTER: Policy Trajectory Ensemble Regularization for Unsupervised Reinforcement Learning"
_TMLR — Accepted by TMLR_

### Review · Reviewer_xtTv · 2023-01-09

**Summary Of Contributions:**

This paper addresses policy pre-training through unsupervised RL. Especially, it proposes a novel regularization method (POLTER) that can be combined with any unsupervised RL algorithm. The regularizer pushes the pre-trained policy closer to a uniform mixture of the previous policies. The paper first motivates the regularization term through the geometric framework of Eysenbach et al. (2022), then provide an illustrative validation in point-mass and pendulum domains, and finally empirically evaluate POLTER in combination of various unsupervised RL methods over the URLB tasks.

**Audience:**

Yes

**Broader Impact Concerns:**

I believe this paper can be labelled as fundamental research, and thus I think it does not need to address specific ethical implications in a Broader Impact Statement.

**Claims And Evidence:**

No

**Requested Changes:**

(Motivation) The paper seems to rely too much on the derivations by Eysenbach et al. (2022) to motivate the proposed regularizer. The adaptation objective in equation (1) is not completely clear to me (what is the policy $\pi$ inducing distribution $\rho (s)$? $\pi_T^*$ is the pre-trained policy or the policy after fine-tuning? Which are the actual optimization variables?), and I have some doubts over the generality of the claim "The optimal prior policy is the one inducing the average state marginal distribution of the skills" (which skills?). Thus, I suggest the authors to expand Section 3.1 to include a more formal motivation of the regularizer and adaptation objective.

(Regularization improves stability) It is known that knowledge-based and data-based methods can suffer from stability issues, as they are usually optimizing an ever changing reward, which sometimes exacerbates the inherent instability of value predictions. The proposed regularizer seems to help the general stability of those methods, as it provides a penalty for deviating from the previous policies. Thus, I am wondering whether the actual motivation of the benefits guaranteed by POLTER is the result of an improved stability of the pre-training.

(State-based evaluation) Whereas the choice to evaluate the proposed regularizer in the state-based version of the URLB is somewhat reasonable, it is not straightforward to say whether POLTER would help in pixel-based tasks as well. The authors are rightly saying that POLTER can be combined with pixel-based methods, but Laskin et al. (2021) show that the performance of the algorithms can change significantly in the two settings (see Figure 3 in Laskin et al.).

(Uniform mixture) The authors are claiming that the theoretical framework in (Eysenbach et al., 2022) supports the use of a uniform mixture of the previous policies. However, it would be interesting to see whether a different choice of $\tilde{\pi}$ can provide even better results. Did the authors tried with an exponentially-decaying mixture that keeps the current policy close to the recent policies?

(Related work) The quickly-growing unsupervised RL literature is now including numbers of methods in the knowledge, data, competence-based categories. The paper is only citing the few works that are most relevant for the subsequent empirical analysis, but I believe the related work section could at least mention the previous literature.

**Strengths And Weaknesses:**

Strengths
- The proposed regularizer is simple to implement and can be easily plugged into various unsupervised RL methods;
- The empirical analysis shows that POLTER improves the performance of some data-based and knowledge-based methods, while it never hurts performance in the presented tasks.

Weaknesses
- The motivation for the proposed regularizer is somewhat weak: The paper does not clearly describe why POLTER should benefit the pre-training performance in general.

This paper is tackling a relevant problem, which is how to pre-train a policy with unsupervised interactions such that to improve the sample-efficiency of learning downstream tasks. In this context, the proposed regularizer seems to provide interesting benefits with little implementation and computation overhead. However, the current version of the paper fails to provide a full understanding on why the regularization actually improves performance of the tested methods, which makes unclear whether we can expect similar benefits in different domains. I report below some more detailed comments and suggestions to improve the paper.

---

> ### Author Response · Authors · 2023-02-23
> **Initial Response**
>
> We want to thank you for the fast and insightful response and for noting the contributions that our paper provides to the field. In the following, we want to address the concerns and describe the changes that we applied to the text.
>
> ## Motivation
> Since this was your main point, we put much effort into expanding the formal derivation of our method in Section 3.1 and Section 3.2.
> The adaptation objective by Eysenbach et al. is a problem statement of Unsupervised Reinforcement Learning (URL), i.e. it describes the abstract objective that an URL algorithm is trying to optimize. The optimization variable is the pretraining prior $\pi(s)$, i.e. the policy after pretraining that minimizes the information cost. In this setting, the algorithm has no direct influence on the worst-case regret as the oracle and task are unknown.
> We agree that the average policy is not the optimal prior in general. This is only true for algorithms that maximize the mutual information $I(\rho(s);z)$ between the state-marginal $\rho(s)$ and a skill $z$. A skill is a representation of a state-dependent reward function that can be defined in the given domain. We expanded the theoretical derivation of the results by Eysenbach et al. in Section 3.1 to show that the average policy over all latent skills is also the optimal prior for data- and knowledge-based URL algorithms.
> We hope that our changes improve the clarity of our motivation for POLTER. Thank you for your questions that have considerably improved the presentation of our method.
>
> ## Regularization improves stability
> We agree that data- and knowledge-based URL algorithms can be unstable due to the changing reward functions that they are trying to optimize. However, applying POLTER has no significant effect on the value predictions or their stability in our experiments. POLTER does indeed slightly reduce the state-marginal entropy, as described in Appendix C, but we do not think that this is the cause for the improved performance. Instead, we see the reduced distance to the optimal prior and thus, the reduced information cost (Equation 1) as the main benefit of POLTER.
> We noticed that the relative improvement of POLTER over the baselines increases with an increase in the number of pretraining steps. So in the new results in Section 5 we analyzed the gradient variance over the course of the pretraining as a measure of how much the policy is changing. Our results show that POLTER prevents the gradient variance from dropping multiple orders of magnitude which we attribute to the more difficult task of simultaneously optimizing the intrinsic reward and minimizing the distance to the ensemble policy (our proxy for the optimal prior).
>
> ## State-based evaluation
> In our original submission, we noted in the limitations that POLTER can readily be applied to environments with pixel-based observations due to a small number of preliminary experiments. We now evaluated POLTER for algorithms from each category (ProtoRL, RND and CIC) on the pixel-based URLB and added the results to Section 5 (Q6) of the paper. They support our initial claim as the improvements of POLTER in IQM are on par with the state-based evaluations. Note that these experiments have a much higher requirement on the compute and memory, so we did not perform the whole set of experiments for all other algorithms.
>
> ## Uniform mixture
> We were also thinking about this question and left a hint in limitations section of our original submission. The problem of finding good mixture coefficients is bound to the structure of the trajectory in policy space during pretraining. If the policy space is perfectly explored, then the uniform mixture is optimal following our assumptions in Section 3.1. Since this ideal case is not reached in our experiments, changing the mixture distribution could have a positive impact on the performance. We tried the suggestion of the reviewer with the exponentially decaying mixture for ProtoRL+POLTER and got to an IQM of 0.72 versus the IQM of the uniform mixture of 0.77. We added this result to our discussion in Section 6.
> However, tuning this hyperparameter is an interesting research direction that requires new insights into the pretraining process and policy trajectory.
>
> ## Related work
> We agree that our paper should be put in context of the constantly growing related work in the field of URL. For this, we added the most relevant papers from this relatively young field to Section 2.2.

---

> > ### Comment · Reviewer_xtTv · 2023-03-01
> > **After Response**
> >
> > I want to thank the authors for their clarifying response and for amending the manuscript to accomodate reviewers' suggestions. Although I think the paper has been improved (especially with pixel-based experiments and additional ablations), I still have doubts on the "optimal prior policy" motivation (more on this below). However, POLTER is essentially a regularization method, and I do not think it is necessary to show it is optimal in some sense as long as it provides clear benefits to the pre-training stability. The latter conclusion seems to be supported by the empirical validation, but not from a complete understanding of the underlying machinery.
> >
> > I report below some follow-up comments.
> >
> > (Motivation) The authors are claiming that the optimal prior policy is the one inducing the average state distribution over all the skills. On the one hand, I think they are only showing that this is optimal to minimize the worst-case information cost, but the analysis does not consider the fine-tuning regret, which also matters. On the other hand, I believe the result in Eysenbach et al. refers to the average of the skills intended as all the vertices of the state distribution polytope. They also show that mutual information maximizing algorithm would not induce all the vertices, and I especially do not see the relation between those vertices and the policies learned while running the URL base algorithm.
> >
> > (Regularization) I think POLTER brings stability to the objective function, which results in improved stability in the value prediction as well. I still do not see how to rule out this motivation to explain the benefit of POLTER.
> >
> > (Related work) I think the related work section is now more complete. Perhaps, a few additional pointers can be found in Laskin et al. (2021). For the data-based method, the authors can also consider mentioning
> > - Seo et al., State entropy maximization with random encoders for efficient exploration, 2021;
> > - Zhang et al., Exploration by maximizing Rényi entropy for reward-free rl framework, 2021;
> > - Mutti et al., Task-agnostic exploration via policy gradient of a non-parametric state entropy estimate, 2021;
> > - Guo et al., Geometric entropic exploration, 2021;
> > - Mutti et al., Unsupervised reinforcement learning in multiple environments, 2022;
> > - Nedergaard and Cook, k-means maximum entropy exploration, 2022.

---

> > > ### Author Response · Authors · 2023-03-07
> > > **Response to Remaining Points**
> > >
> > > Thank you for engaging in this discussion and appreciating the improvements we have applied to the manuscript since the initial submission. In the following, we will address your remaining points.
> > >
> > > ## Motivation
> > >
> > > You are correct in that the average policy over all skills only minimizes the worst-case information cost. We will make this distinction more clear in the next revision. The question of how to take the finetuning regret into account leads in the direction of meta RL, of which we see unsupervised RL as a special case.
> > >
> > > The relation between the vertices of the state-marginal polytope and the policies that are learned by the URL algorithm during pretraining is based on Equation 3 and our Lemma 2. An algorithm that maximizes the mutual information between the states and a skill will induce a state-marginal that lies on a vertex of the state-marginal polytope (Lemma 6.1 in Eysenbach et al.). As Eysenbach et al. showed, they won’t generally cover all vertices, and the research on algorithms that are able to cover all vertices of the state-marginal polytope is still an open question.
> > >
> > > The skills of data- and knowledge-based URL algorithms are encoded by the intrinsic reward objective, such as, e.g., the random network distillation error (RND) or the entropy of the prototypical state-marginal (ProtoRL). If we fix the intrinsic reward mechanism at a fixed point during pretraining and let an URL algorithm such as ProtoRL optimize this objective, it will maximize the entropy of the state-marginal and find the optimal policy for this fixed latent skill. Using our Lemma 2, this will also minimize the conditional entropy of the state-marginal given the skill (Equation 3). Thus, it will maximize the mutual information between the state-marginal and the fixed skill and will lie on some vertex of the state-marginal polytope so that the results of Eysenbach et al. are also applicable in this case.
> > >
> > > Note, however, that this is the ideal case where the URL algorithm is able to optimize each latent skill (state of the intrinsic return mechanism) until it finds its optimal policy. Our heuristic for the ensemble member checkpoints $\mathcal{T}_E$ is based on the changes in the intrinsic reward signal that indicate a shift in the latent skill, but a principled method to infer this change is a possible direction for future work. Additionally, the skills that are encoded in the intrinsic return objective are not independent of each other and might be too similar so that their state-marginals will all lie on the same vertex. This happens if the exploration of the URL algorithm during pretraining does not work well, as is the case for Disagreement in our experiments in Section 5. Tackling this problem is another interesting research direction.
> > >
> > > ## Regularization
> > >
> > > We analyzed the value losses and predictions of ProtoRL with and without POLTER and found that POLTER generally increases their variance and magnitude throughout the pretraining phase. This is probably due to the algorithm trying to simultaneously fit the value function and match the ensemble policy and is reflected in the gradient variance experiment that we added to Section 5. This led us to conclude that the POLTER regularization results in a better initialization for the downstream tasks instead of a more	 stable value prediction.
> > >
> > > ## Related Work
> > >
> > > We agree that the papers that you listed should be included as pointers in the related work section and will add them to the next revision. Thank you for providing these to more thoroughly embed our work in the related context.

---

> > > > ### Comment · Reviewer_xtTv · 2023-03-08
> > > > **Thanks for Clarifications**
> > > >
> > > > I thank the authors for their further clarifications.
> > > >
> > > > Now I believe to have a clearer understanding of what was the ideal objective for POLTER, and the sequence of approximations that led to the presented heuristic. On the one hand, it is unclear whether minimizing the worst-case information cost is the "right" objective for unsupervised RL, although it appeared in the literature before (mostly in Eysenbach et al.). On the other hand, without any link between the checkpoints and the state of the underlying learning process (i.e., actually looking for when the base algorithm is converging to a vertex) is hard to see whether POLTER is really optimizing this objective in practice.
> > > >
> > > > Anyway, I still see value in this paper, but I will provide a borderline evaluation in consideration of my concerns on the motivation.

---

> > > > > ### Author Response · Authors · 2023-03-08
> > > > > **Regarding the Objective of Unsupervised RL**
> > > > >
> > > > > Thank you again for this active discussion. Developing a method to analyze the structure of the state-marginal space and to check whether the current state-marginal of the Unsupervised RL (URL) algorithm has converged to a vertex is definitely a valuable question for future work. Currently, we are unaware of any such algorithm but we see the results of our heuristic as an indicator of the merits that our method can bring.
> > > > >
> > > > > Whether minimizing the worst-case information cost is the “right” goal for URL is a question that is worthwhile to discuss. The formulation by Eysenbach et al. can be related to the framework of information-theoretic bounded rationality [1]. In this case, the utility function $U$ is the negative regret on an adversarially chosen downstream task and the goal of URL is the minimization of the *free energy* functional of the RL agent’s policy (see Equation 6 in [1]).
> > > > > Other utility functions could incorporate knowledge about the structure of the downstream task that can in turn be used to guide the pretraining phase. We see this question as a great direction for future research.
> > > > >
> > > > > [1] Ortega and Lee, “An Adversarial Interpretation of Information-Theoretic Bounded Rationality”, 2014, https://ojs.aaai.org/index.php/AAAI/article/view/9071/8930

---

### Review · Reviewer_tT3Y · 2023-02-03

**Summary Of Contributions:**

This paper presents Policy Trajectory Ensemble Regularization (POLTER), a method to improve the performance of unsupervised RL algorithms. POLTER introduces a KL regularization term between the learning policy and the ensemble of pretraining policies in the policy learning objective function. In the PointMass domain, they showed that POLTER obtains the lower KL-divergence to the optimal pretraining policy. In the experiments, POLTER is shown to be effective to improve the existing URL algorithms, especially data-based and knowledge-based algorithms.


**Audience:**

Yes

**Claims And Evidence:**

Yes

**Requested Changes:**

It would be great to provide (1) a theoretical justification for POLTER. (2) More baseline experimental results for PointMass/Pendulum (currently only RND is being considered). (3) Explanation of how to choose the checkpoint intervals for the ensemble policy and experimental results on how sensitive POLTER's performance is depending on this choice.

**Strengths And Weaknesses:**

[Strengths]
- The proposed POLTER is very simple to implement, which only requires storing pertaining policy checkpoints for an ensemble policy and adding KL-divergence regularization term between the ensemble policy and the learning policy to the objective function.
- It is empirically shown that POLTER archives the policy that yields a smaller divergence to the optimal prior policy in the PointMass domain, which coincides with the main motivation of the paper.
- In the URLB benchmark, POLTER generally improves the performance of the existing baseline algorithms, especially for data-based and knowledge-based methods.
- Various ablation experiments are provided, which is great for a better understanding of when/how POLTER works well or not.

[Weaknesses]
- Theoretical analysis is lacking. Beyond the intuition level, it is unclear how minimizing the KL divergence to the ensemble policy can lead to an optimal prior policy. Since the policy and its state occupancy measure have a non-linear relationship, it is unclear whether an average of multiple policies induces a state-marginal distribution defined in terms of a convex combination of each policy's state marginal. It would be great to provide some theoretical justification for the provided method.
- In Figure 3, only RND results are provided. Besides RND, it would be great to see the results of other URL baselines: whether adding POLTER obtains a resulting policy that is consistently closer to the optimal prior policy.
- In addition to $\alpha$, POLTER has an additional hyperparameter, $\mathcal{T}_E$. How often should we add policy checkpoints to the ensemble, and how sensitive is the performance of POLTER depending on this choice?

---

> ### Author Response · Authors · 2023-02-23
> **Initial Response**
>
> Thank you for the detailed analysis of our method and the positive reception of our work. In the following, we want to address the points that you raised for improving the presentation and evaluation of our method.
>
> ## Theoretical Justification
>
> We expanded the Sections 3.1 and 3.2 to provide a more formal justification of applying the results by Eysenbach et al., which were originally developed for mutual information maximizing skill-based algorithms, to data- and knowledge-based URL algorithms. We also added a clarifying note regarding your question of the connection between the ensemble mixture policy and the average state-marginal. Our argument is the same as that of Eysenbach et al. in their Appendix A.2, as we would apply our non-markovian policy by sampling the index of the mixture component at the start of each episode and then following this policy afterwards. Because the policies in our case are truncated gaussians, our mixture policy is a gaussian mixture model with components $\frac{1}{k}$, where $k$ is the size of the ensemble. We added this explanation to Section 3.2.
>
> ## Demonstration on PointMass
>
> We agree with you that the results on PointMass should also include other URL algorithms to show the generality of our method. For this we evaluated ProtoRL and added the results to Appendix A.
> For Pendulum, the results for RND, ProtoRL and CIC can be found in Appendix B.
>
> ## Sensitivity Analysis for $\mathcal{T}_E$
>
> In initial experiments, we set $\mathcal{T}_E$ to align with the changes in the intrinsic reward changes during pretraining of RND on the Quadruped domain, as described in Appendix F. We wanted to keep the amount of tuning this hyperparameter to a minimum, as it goes against the goal of unsupervised RL to increase the sample-efficiency. In practice, having to train the unregularized variant of an URL algorithm to choose this hyperparameter would almost double the amount of environment interactions. However, we agree with you that a principled analysis of this hyperparameter would be insightful for the inner workings of our method. We added an analysis of several different checkpoint schedules to Section 5. We chose three simple alternative schedules with checkpoints added every 50k, 100k and 200k steps to show the variation in performance.

---

> > ### Comment · Reviewer_tT3Y · 2023-03-23
> > **Thanks**
> >
> > I would like to thank the authors for their response. I have read the revised manuscript, where additional justification and experiments are greatly appreciated. I believe most of my concerns have been addressed.

---

### Review · Reviewer_hyqK · 2023-02-14

**Summary Of Contributions:**

This paper proposes a regularization technique for Unsupervised Reinforcement Learning (URL) which involves minimizing the KL divergence to an ensemble of policies discovered during the pre-training process. This approach can be easily applied to large set of URL algorithms and is shown to improve the performance of a set of data and knowledge-based URL algorithms on the URL benchmarks.

**Audience:**

Yes

**Claims And Evidence:**

Yes

**Requested Changes:**

See previous section

**Strengths And Weaknesses:**

Overall I quite like this paper, the proposed technique is well-grounded theoretically and is shown to have good empirical results. I especially like the simplicity of the idea and can see this as a new standard addition to future URL algorithms, my comments are as follows:
- The proposed method is intuitive and has theoretical support from previous work. The idea that each point during pre-training encodes an implicit skill, averaging these results can result in a proxy to the average state marginal of skills is a very natural and direct application of the insights from Eysenbach et al. (2022).
- The proposed approach is straightforward to implement and widely applicable to a whole range of URL algorithms.
- I believe the main strength of this paper is its experiments. The set of experiments is quite thorough and provides an abundance of empirical evidence of both the effectiveness and wide-adaptability of the technique. The authors also did a good job of analyzing their approach for different categories and sensitivity to hyperparameters. I also really like the discussion on state-visitation entropy in Appendix C which provides empirical evidence on the relationship between space space exploration on POLTER.
- While the paper is mostly clear, I do think some additional clarifications could make the paper more self-contained. For example, some concrete examples/descriptions on data/knowledge/skill-based algorithms and how POLTER works in each case would help visualize the author's ideas better.
- I recommend defining $\pi(s)$ in section 3.2 and adding some clarification on what is meant by an "optimal oracle policy", this relates back to the point of making the paper more self-contained.
- I think it is fair to present the results for an $\alpha$ tuned for each domain and task since the authors did explicitly note this and that the authors did also present results of the untuned variant. However, I think the way the term "tuned variants" is used on page 10 might be a little misleading as it seems imply some kind of auto-tuning approach to finding $\alpha$.
- Not asking this as a requested change but related to the previous point, do you think there is any way to adaptively tune $\alpha$?
- Could you elaborate on some of the challenges and difficulties of extending POLTER to pixel-based environments? On a high level, what modifications do you envision you have to make?

---

> ### Author Response · Authors · 2023-02-23
> **Initial Response**
>
> We appreciate your positive reception of our work, especially for noting the simplicity of the method, and will address your questions in the following paragraphs.
>
> ## Theoretical Derivation
>
> We substantially extended the theoretical derivation of POLTER in Sections 3.1 and 3.2 and relate it to the different types (knowledge/data/skill-based) of URL algorithms. Regarding the *optimal oracle policy*: This is the (generally unknown) policy that optimally solves the worst-case downstream task. We reformulated this in the main text.
>
> ## Tuning-Terminology
>
> We agree with you with the ambiguity of the tuning term, we added that we used a grid sweep for tuning in the text.
>
> ## Adaptive $\alpha$
>
> When we talk about tuning $\alpha$ and adaptively setting $\alpha$ we differentiate two methods.
> The first is finding one static $\alpha$ which we keep across the whole pretraining run.
> For this task, there exist different approaches and established tools like grid sweeps, random search or Bayesian Optimization methods (e.g., SMAC3 [Lindauer et al., 2022]).
> With prior knowledge about the downstream tasks this would fall into the category of algorithm configuration.
> Whereas with adaptively adjusting $\alpha$ we could hope to adjust the strength of the regularization during the run. As (U)RL is an iterative learning process it is possible that the required strength of the regularization might change, as this has been observed with other hyperparameters in other domains [Karafotias et al., 2015; Doerr and Doerr, 2020; Speck et al., 2021; Adriaensen et al., 2022].
> Discovering and defining dynamic hyperparameter schedules is far from solved, and a computationally intensive problem like this poses an additional challenge to developing such methods.
> One could start with simple heuristics derived from domain knowledge up to a data-based approach.
> It is very probable that the different domains/environments require different $\alpha$-schedules as different RL algorithms and environments require different hyperparameters.
>
>
> ## Pixel-Based Observations
>
> Because of the simplicity of our regularization POLTER, we do not need to change anything algorithm-wise to also use it on pixel-based observations.
> After submitting the initial version of the paper, we were able to evaluate POLTER for CIC, RND and ProtoRL on the URLB with pixel-based observations and observed the same results as for the state-based evaluation. We added the results to Section 5.

---

> > ### Comment · Reviewer_hyqK · 2023-03-19
> > **Response**
> >
> > I want to thank the authors for addressing my comments and concerns, I have read the revised version of the paper and I am happy with your response and revision.

---

### Decision · Action_Editors · 2023-04-09

**Recommendation:** Accept with minor revision

**Comment:**

Reviewers find the work to be useful contributions about Unsupervised Reinforcement Learning, with good supports. The revised manuscript addresses earlier feedback with a number of improvements. We recommended acceptance, with a few minor suggestions for the final version:

* page 4: in equation 2, the domain Z is still unclear. Is it over all (infinitely many) possible skills, or a finite set of skills? How is it determined (human-defined, or automatically discovered, or else)? A sentence or two will make the paper more self-contained.
* page 4: typo “an dynamics model”
* page 5: mark the end of Proposition 1’s proof sketch.
* The notation of mutual information is convenient, but not standard (in my opinion). MI is a function of two random variables, not of a (distribution, random variable) pair. If the authors prefer to use the current notation, please add a sentence to explain the choice. It also avoids confusion (e.g., when readers refer to the relevant papers like Eysenbach et al. that uses the standard notation).



**Audience:**

The work is interesting to a community in unsupervised reinforcement learning, and potentially to a broader community in AI that works on model pre-training.

**Claims And Evidence:**

The claims are supported by theory (proposition 1) and experiments.